# Learning Deformable Body Interactions With Adaptive Spatial Tokenization

**Hao Wang**[*]
**Apple**
devin.wanghao@apple.com

**Yu Liu**[*]
**Apple**
yliu66@apple.com

**Daniel Biggs**
**Apple**
dbiggs@apple.com

**Haoru Wang**
**Apple**
haoru_wang@apple.com

**Jiandong Yu**
**Apple**
jiandong_yu@apple.com

**Ping Huang**
**Apple**
huang_ping@apple.com

**Reviewed on OpenReview:** https://openreview.net/forum?id=qBOEHzkr1P

## Abstract

Simulating interactions between deformable bodies is vital in fields like material science, mechanical design, and robotics. While learning-based methods with Graph Neural Networks (GNNs) are effective at solving complex physical systems, they encounter scalability issues when modeling deformable body interactions. To model interactions between objects, pairwise global edges have to be created dynamically, which is computationally intensive and impractical for large-scale meshes. To overcome these challenges, drawing on insights from geometric representations, we propose an *Adaptive Spatial Tokenization (AST)* method for efficient representation of physical states. By dividing the simulation space into a grid of cells and mapping unstructured meshes onto this structured grid, our approach naturally groups adjacent mesh nodes. We then apply a cross-attention module to map the sparse cells into a compact, fixed-length embedding, serving as tokens for the entire physical state. Self-attention modules are employed to predict the next state over these tokens in latent space. This framework leverages the efficiency of tokenization and the expressive power of attention mechanisms to achieve accurate and scalable simulation results. Extensive experiments demonstrate that our method significantly outperforms state-of-the-art methods in modeling deformable body interactions. Notably, it remains effective on large-scale simulations with meshes exceeding *100,000 nodes*, where existing methods are hindered by computational limitations. Additionally, we contribute a novel large-scale dataset encompassing a wide range of deformable body interactions to support future research in this area.

## 1 Introduction

Solving interactions between deformable bodies plays a vital role in a wide range of applications, including material science (David Müzel et al., 2020; Barbero, 2023), mechanical design (Thompson & Sung, 1986; Zienkiewicz & Taylor, 2005), and robotics (Collins et al., 2021). The finite element method (FEM) is a widely used numerical approach for addressing such problems (Courant et al., 1994). However, FEM typically incurs high computational costs and requires significant manual effort from engineers to ensure solver stability and convergence (Oc, 2000; Hughes, 2003; Cook et al., 2007; Elrefaie et al., 2024). Recently, there has been growing interest in leveraging learning-based methods to address deformable body simulation. MeshGraphNet (MGN) and related approaches (Pfaff et al., 2020; Sanchez-Gonzalez et al., 2020; Fortunato et al., 2022) represent unstructured meshes as graphs and employ stacked message-passing blocks to propagate physical information across the mesh. Subsequent variants have introduced various graph pooling operations and

---

[*]These authors contributed equally.

U-Net-like architectures (Lino et al., 2021; Deshpande et al., 2024; Cao et al., 2023) to solve the multi-scale challenges in different simulation tasks. To model interactions between distinct deformable bodies, these methods typically construct dynamic edges based on the proximity of mesh nodes (Pfaff et al., 2020; Yu et al., 2024). However, maintaining global pairwise edges across all mesh nodes becomes a significant computational bottleneck, limiting the scalability of such models to larger or more complex scenes. Rubanova et al. (2024) proposed to use 3D implicit representations like Signed Distance Function (SDF) for more efficient collision detection, but it has to be built on the solid body prior. Thuerey et al. (2018) interpolated the input 2D field onto a 128×128 grid and applied a convolutional U-Net to process the grid input for CFD simulation. O Pinheiro et al. (2023) proposed representing atoms as continuous densities and molecules as discretizations of 3D space on voxel grids. Similar to image tasks, they applied a score-based generative model to generate molecules by denoising noisy voxelized representations. Beyond regular grids suitable for convolutional operations, later methods such as GraphCast (Lam et al., 2023) introduced a background grid to which the mesh is attached, enabling more efficient message passing in graph neural networks. However, this effectively converts an unstructured mesh into a structured one, which may work well for simple, regular geometries but can suffer from reduced accuracy and efficiency when applied to complex shapes.

Methods such as GraphCast (Lam et al., 2023) propose creating a background grid to which the mesh is attached, enabling more efficient message passing. However, this effectively converts an unstructured mesh into a structured one, which can work well for simple, regular geometries such as spheres. When applied to complex shapes, however, this approach may suffer from reduced accuracy and efficiency. Transformer-based models such as HCMT (Yu et al., 2024) propose using two sets of attention blocks to model contacts and collisions between deformable objects. However, the attention matrices are constructed over mesh nodes and global edges, resulting in high memory consumption. This makes the approach computationally prohibitive when scaling to larger meshes.

Recent advances in computer vision (Dosovitskiy et al., 2020; Radford et al., 2021; Li et al., 2023) and computer geometry (Zhang et al., 2023b; Li et al., 2025) have shown that applying learnable tokenization to raw input signals is critical for downstream understanding and generation tasks. In computer vision (Dosovitskiy et al., 2020; Radford et al., 2021), input images are typically divided into patches, which are then mapped via linear layers to patch embeddings. These embeddings, along with added positional information, are processed further using token-wise operations. Similarly, in geometric tasks, 3D shapes—represented as meshes, point clouds, or signed distance functions (SDFs)—are typically embedded into compact latent representations before being passed to downstream components (Zhang et al., 2023a; Li et al., 2025). These approaches inspired us to design an effective tokenization strategy for physical states in simulation, where the state can be viewed as a set of vector fields defined over a given 3D geometry.

In this work, we introduce **Adaptive Spatial Tokenization (AST)**, a novel method for encoding diverse physical states into fixed-length embeddings. We begin by quantizing the spatial domain into a grid of cells, effectively mapping the unstructured mesh onto a structured spatial partition. To manage this representation efficiently, we also provide the option to use an octree-like hierarchical indexing system for scalable storage and fast lookup. As showed in Figure 1, in our proposed representation, adjacent mesh nodes are naturally grouped into shared cells, enabling structured local interactions. We then apply sparse convolution to propagate information between cells, allowing inter-object interactions to emerge naturally. This approach eliminates the need for explicitly constructing costly pairwise global edges, which is a common bottleneck in prior graph-based methods. Inspired by the use of cross-attention in Transformer architectures and its success in 3D shape representation(Zhang et al., 2023b; Li et al., 2025), we design a cross-attention module that queries features from the quantized sparse cells using compact, fixed-length vectors—our adaptive spatial tokens. These tokens are then processed in latent space using a Transformer-style network composed of a series of self-attention layers, enabling next-step prediction of the physical state. We conduct extensive experiments across various scenarios involving deformable body interactions. Our method consistently outperforms state-of-the-art baselines, particularly in modeling inter-object interactions. Furthermore, in large-scale simulations involving meshes with over **100,000 nodes**—where most existing methods fail due to memory constraints—our approach continues to produce accurate and stable predictions.

Our primary contributions can be summarized as follows:

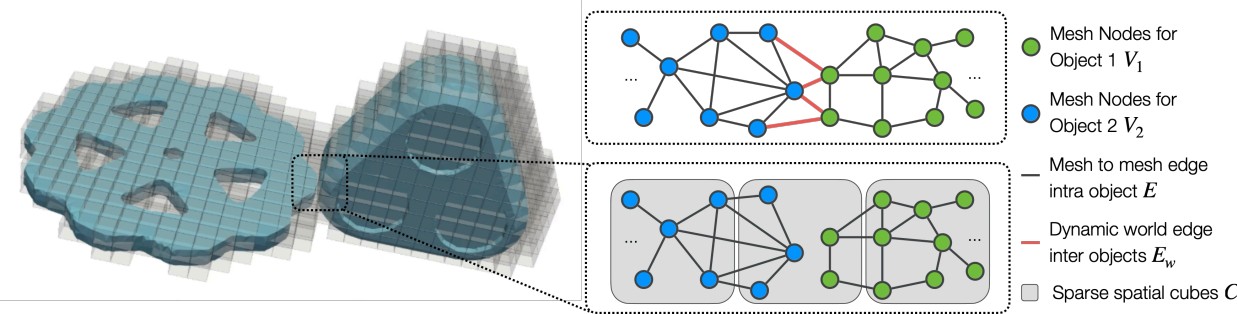

Figure 1: Spatial Cells for interactions.

1. We propose Adaptive Spatial Tokenization (AST) to efficiently represent physical states in simulation by mapping unstructured meshes into structured grid cells and compressing them into a compact, fixed-length representation via cross-attention.

2. We develop an attention-based model that operates on spatial tokens in latent space to simulate interactions between deformable bodies, without relying on explicit global connectivity.

3. Our method achieves significant improvements over state-of-the-art baselines, particularly in modeling complex object interactions.

4. We introduce ABCD-XL, a novel large-scale dataset featuring diverse deformable body interactions for benchmarking.

## 2 Related Work

In recent years, solving physics simulation with learning based methods have been an active research area. One of the most representative work is MGN (Pfaff et al., 2020), in which mesh data with physical states is treated as graph. It adopts Encode-Process-Decode architecture (Sanchez-Gonzalez et al., 2020), stacking multiple message passing layers as the core processing module to propagate the physics information through the graph. Several variants, such as BSMS (Cao et al., 2023), MAgNET (Deshpande et al., 2024), and GNN-U-Net (Gladstone et al., 2023), introduce enhancements including virtual edges, graph pooling, and U-Net structures to improve computational efficiency and handle long-range interactions. To simulate interactions between objects, these methods typically construct dynamic world edges by connecting spatially close mesh nodes across objects at each time step. FIGNet (Allen et al., 2023) extends this idea by defining multiple edge types (face-to-face, mesh-to-mesh, and object-to-mesh) to better capture rigid body dynamics. More recently, SDF-Sim (Rubanova et al., 2024) introduces implicit representations such as signed distance functions (SDFs) for efficient collision detection between rigid bodies. However, both methods focus on rigid body interactions and do not offer scalable solutions for deformable body simulations. HCMT (Yu et al., 2024) explores the use of Transformer-style attention blocks to model the dynamics between deformable bodies. While promising, it still relies on constructing world edges and requires computing dense node-wise attention matrices, which limits its scalability to large meshes.

Recently, learnable tokenizers have been widely adopted in both computer vision (Oord et al., 2017; Dosovitskiy et al., 2020; Radford et al., 2021) and computer geometry (Wang, 2023; Zhang et al., 2023b; Li et al., 2025) to improve efficiency and scalability, leading to state-of-the-art performance in downstream tasks such as understanding and generation. In vision tasks, methods like ViT (Dosovitskiy et al., 2020) first split the input image into fixed-size patches. These patches are mapped to embeddings using linear layers, which, when combined with positional embeddings, form patch-level image tokens. VQ-VAE (Oord et al., 2017) uses a variational autoencoder to learn reconstructable tokens for image patches and applies vector quantization to map them into a discrete codebook. In computer geometry, hierarchical structures like the Octree (Meagher, 1982) are designed for efficient storage of 3D shapes and their properties (Wang et al., 2017). In Octformer (Wang, 2023), following the approach of ViT, sparse convolutions are applied to structured

point clouds to tokenize the input and map it into latent space for downstream tasks. Shape2VecSet (Zhang et al., 2023a) and Tripos (Li et al., 2025) use cross-attention modules to map 3D shapes to fixed-length vectors, treating them as latent tokens. Diffusion models are then trained over these learned tokens for generation tasks.

Building on insights from previous works, we treat the physical state in simulations as vector fields defined over a given 3D shape, and propose Adaptive Spatial Tokenization to push the boundaries. Inspired by approaches in graphics and vision tasks, our pipeline divide vectors in the space to cell patches, encode cells to compact, fixed-length tokens, and apply attention-based modules to complete next-step prediction in the simulation. We conduct a series of experiments and ablation studies to demonstrate how Adaptive Spatial Tokenization enhances both the efficiency and performance of simulations involving deformable bodies with interactions.

## 3 Preliminaries

In this section, we briefly introduce the preliminary techniques used in our method and refer interested readers to the original sources for further details.

### 3.1 Graph Operations

Graphs offer a flexible structure for representing complex data. We leverage message-passing operations to aggregate features on input graphs before transforming them into discrete tokens.

**Message-Passing** Message passing refers to feature aggregation operations over graphs. While numerous variants exist in the literature, we adopt the formulation from Pfaff et al. (2020) and extend it to heterogeneous graphs. Given an edge set $E_t$, edge attributes $\{e_{ij}\}$ defined on each edge, and the corresponding sender and receiver node features $\{v_i^s\}$ and $\{v_j^r\}$, the message passing operation $v^r \leftarrow \text{Message-Passing}(e, v^s, v^r)$ proceeds in two steps:

$$
\begin{aligned}
e'_{ij} &= f^e(e_{ij}, v_i^s, v_j^r), \\
v'^r_j &= f^v(v_j^r, \sum_i e'_{ij}),
\end{aligned}
\tag{1}
$$

where $f^e$ is the edge update function and $f^v$ is the node update function, typically implemented as multilayer perceptrons (MLPs). The implementation details of the MLPs can be found in Section A.5. The sum is taken over all sender nodes $i$ connected to receiver node $j$. Message passing operations can also be applied in the absence of explicit edge features, resulting in an edge-free formulation: $v^r \leftarrow \text{Message-Passing}(v^s, v^r)$, where the edge update function simplifies to:

$$
e'_{ij} = f^e(v_i^s, v_j^r).
\tag{2}
$$

### 3.2 Spatial Operations

Although graph operations are flexible and generalizable to various data structures, they can be inefficient and memory-intensive due to explicit edge representations. To address this, we introduce spatial techniques from the 3D vision literature—originally designed for large-scale point clouds—to enable more efficient computation.

**Sparse Convolution** Sparse convolution is a powerful technique widely used in 3D shape analysis and synthesis (Graham, 2015; Yan et al., 2018; Wang et al., 2017). It enables efficient operations on sparse 3D representations, such as point clouds and volumetric grids, by leveraging octree structures to compress spatial information without loss. In the context of mesh-based physical simulation, we briefly outline how sparse convolution is applied, and refer readers to Wang et al. (2017) for more comprehensive details.

Starting from a unit cell (the level-0 cell defined over the space $[-1, 1]^3$) that encompasses the entire 3D object, an octree is constructed by recursively subdividing each cell into eight child cells whenever the parent cell contains at least one mesh node. This process continues until a maximum predefined level-$L$ is reached,

and the cells at level-$L$ have side length $2^{1-L}$. Each cell is assigned a coordinate $\boldsymbol{P}_l = (x_l, y_l, z_l)$ that indicates its position in the uniform grid at level-$l$, derived by uniformly dividing the unit cell into $2^{1-l}$ segments per axis.

A sparse convolution from level-$(l+s)$ to level-$l$ is applied at each non-empty level-$l$ cell $\boldsymbol{c}_i^l$ using the following rule:

$$\boldsymbol{c}_i^l = \sum_{k=1}^K \boldsymbol{w}_k^l \cdot \boldsymbol{c}_{\mathcal{N}(l+s,i,k)}^{l+s} + \boldsymbol{b}^l, \quad i \in [1, N^l], \tag{3}$$

where $\boldsymbol{c}_i^l$ denotes the feature at $\boldsymbol{c}_i^l$, $N^l$ is the number of non-empty cells at level-$l$ and $\boldsymbol{v}_t'^c = [\boldsymbol{c}_1^L, ..., \boldsymbol{c}_{N^L}^L]$. $K$ is the number of neighbors involved in the convolution, $w_k$ and $b$ are learnable convolution weights and bias, and $\mathcal{N}(l+s, i, k)$ returns the $k$-th neighboring cell relative to $\boldsymbol{c}_i^l$ at level-$(l+s)$. More precisely, based on the cell at position $\boldsymbol{P}_{l+s} = \text{floor}\{\boldsymbol{P}_l/2^s\}$, where $\boldsymbol{P}_l$ is the position of $\boldsymbol{c}_i^l$, $\{\mathcal{N}(l+s, i, k)\}_{k=1}^K$ designates a group of relative positions representing the convolution kernel. For example, a $3 \times 3 \times 3$ convolution kernel corresponds to $\{\mathcal{N}(l+s, i, k)\}_{k=1}^K$ pointing to the cells at positions:

$$\{(x_{l+s} + \alpha, \ y_{l+s} + \beta, \ z_{l+s} + \gamma)\}, \quad \alpha, \beta, \gamma \in \{-1, 0, 1\}. \tag{4}$$

If no cell is present at the corresponding position, the features are treated as all zeros, similar to a padding operation. Recursively applying sparse convolution layers forms an OCNN operation, defined as

$$\text{OCNN}(\boldsymbol{c}^L) = \text{SparseConv}^{N_{\text{conv}}}(...(\text{SparseConv}^1(\boldsymbol{c}^L))), \tag{5}$$

where $N_{conv}$ denotes the number of convolutional layers, and each $\text{SparseConv}^i$ operation can have distinct configurations based on specific design choices.

**Farthest Point Sampling** The Farthest Point Sampling (FPS, (Eldar et al., 1997; Qi et al., 2017)) algorithm selects a representative subset of points (or cells, in our context) based on spatial distribution. It is defined as

$$\boldsymbol{h} = \text{FPS}(\boldsymbol{c}, \boldsymbol{p}^c), \tag{6}$$

where $\boldsymbol{c}$ is the input feature set with corresponding spatial positions $\boldsymbol{p}^c$, and $\boldsymbol{h} \subset \boldsymbol{c}$ is the sampled subset.

# 4 Method

## 4.1 Problem Setup

We consider the evolution of the physics-based system discretized on a mesh, which could be directly represented by a heterogeneous graph $G_t = (\{\mathcal{V}_t^m, \mathcal{V}_t^e\}, \{\mathcal{E}_t^{m2m}, \mathcal{E}_t^{m2e}, \mathcal{E}_t^{e2m}\})$. $\mathcal{V}_t^m$ and $\mathcal{V}_t^e$ represent the node sets associated with physical properties (e.g., material properties, strain, stress) defined on mesh nodes and element nodes, respectively. The edge sets $\mathcal{E}_t^{m2m}$, $\mathcal{E}_t^{m2e}$, and $\mathcal{E}_t^{e2m}$ capture physical relationships between mesh-to-mesh, mesh-to-element, and element-to-mesh pairs, respectively.

We choose to include the element nodes besides the mesh nodes to form a heterograph, as we found that explicitly modeling elements is critical for realistic physical simulations. Many physical quantities—such as strain and stress—are defined via integration over entire elements rather than at individual nodes. Thus, representing such properties at the element level aligns more naturally with formulations found in classical PDE solvers.

At each time step $t$, certain node positions or physical properties may be externally specified. These are collectively referred to as the boundary condition $\mathcal{B}_t$. For example, in a quasi-static scenario where a deformable object is being compressed by a rigid body, the movement of the rigid body must be provided; otherwise, the resulting deformation cannot be inferred solely from the current state.

The objective is to model the evolution of the vector field by learning a transformation $\mathcal{F}$:

$$\hat{G}_{t+1} = \mathcal{F}(\mathcal{B}_t, G_t, G_{t-1}, G_{t-2}, \ldots, G_{t-h+1}), \tag{7}$$

Figure 2: Model structure overview. Graph-based physical states are encoded into latent tokens via Adaptive Spatial Tokenization (AST), processed with attention-based mechanism, and decoded back for next-step prediction. The green cells are selected by the FPS algorithm and serve as query tokens in the cross-attention shown in the top row.

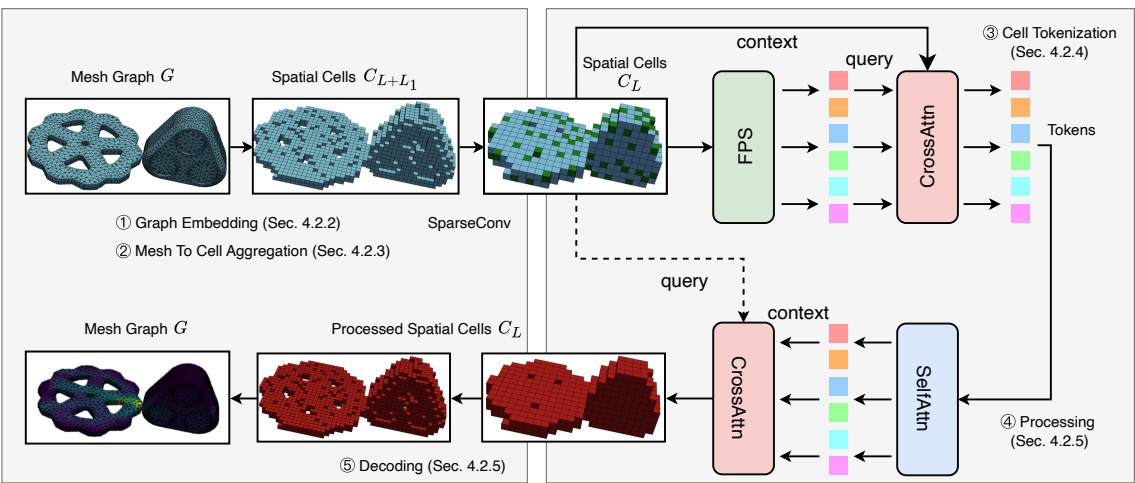

Mesh Aggregation (**Section 4.2.3**)    Cell Tokenization (**Section 4.2.4**)

where $h$ is the number of historical steps considered.

## 4.2 Adaptive Spatial Tokenization

### 4.2.1 Overall Architecture

Existing methods typically employ message-passing or graph pooling operations to retain the expressive power of graph representations. However, graph-based structures do not scale well to large-scale meshes, and their information aggregation is inherently slow due to the localized nature of propagation. To overcome these limitations, our novel approach—Adaptive Spatial Tokenization (AST)—aggregates features defined on graphs into compact latent tokens through the following steps:

1. Encode the raw features on $G_t$ into an embedded feature graph $\bar{G}_t$ (Section 4.2.2).

2. Aggregate the mesh node features $\bar{\mathbf{v}}_t^m$ from $\bar{G}_t$ into cell features $\bar{\mathbf{v}}_t^c$ defined on spatial cells $C_{L,t}$ (Section 4.2.3).

3. Project the cell features $\bar{\mathbf{v}}_t^c$ into a fixed-length set of latent tokens $\mathbf{h}_t$ (Section 4.2.4).

4. After transformer-based processing on $\mathbf{h}_t$, decode the tokens back to spatial cells $C_{L,t}$, and then reconstruct the output graph $\hat{G}_{t+1}$ (Section 4.2.5).

Notably, the spatial cells are constructed on a per-frame basis to capture instantaneous spatial interactions at each time step. The overall model architecture is illustrated in Figure 2.

### 4.2.2 Graph Embedding Encoder

The input heterograph $G_t$ will first be transformed to $\bar{G}_t = (\{\bar{\mathcal{V}}_t^m, \bar{\mathcal{V}}_t^e\}, \{\bar{\mathcal{E}}_t^{m2m}, \bar{\mathcal{E}}_t^{e2m}, \bar{\mathcal{E}}_t^{m2e}\})$ via the graph embedding encoders. Specifically, node features $\mathbf{v}_t^m$, $\mathbf{v}_t^e$ and edge features $\mathbf{e}_t^{m2m}$, $\mathbf{e}_t^{m2e}$, $\mathbf{e}_t^{e2m}$ are projected into latent space using MLPs, resulting in $\bar{\mathbf{v}}_t^m$, $\bar{\mathbf{v}}_t^e$, $\bar{\mathbf{e}}_t^{m2m}$, $\bar{\mathbf{e}}_t^{m2e}$, and $\bar{\mathbf{e}}_t^{e2m}$. An E2M (element-to-mesh)

message-passing operation is then applied to aggregate features from element nodes to mesh nodes:

$$\bar{\mathbf{v}}_t^m \leftarrow \text{Message-Passing}(\bar{\mathbf{e}}_t^{e2m}, \bar{\mathbf{v}}_t^e, \bar{\mathbf{v}}_t^m). \tag{8}$$

Optionally, or when no element-level features are available, an M2M (mesh-to-mesh) message-passing operation can be performed to encode positional and structural information via $E_t^{m2m}$:

$$\bar{\mathbf{v}}_t^m \leftarrow \text{Message-Passing}(\bar{\mathbf{e}}_t^{m2m}, \bar{\mathbf{v}}_t^m, \bar{\mathbf{v}}_t^m). \tag{9}$$

These message-passing operations encode the input graph structure and features into the mesh node representation $\bar{\mathbf{v}}_t^m$.

### 4.2.3 Mesh To Cell Aggregation

Although graphs offer high flexibility and generalization across diverse data structures, message-passing operations typically require explicitly materializing $\mathbf{v}^s$ and $\mathbf{v}^r$ along graph edges, which leads to significant memory overhead on large-scale graphs. Our method overcomes this limitation by partitioning space into a regular grid and mapping mesh nodes onto it.

We construct an octree of depth $L$ based on the mesh node positions $\mathbf{p}_t^m$, and the non-empty leaf cells are referred to as $C_{L,t}$. A visualization is shown in Figure 1. We then establish edge sets between mesh nodes and spatial cells—denoted $\mathcal{E}_t^{m2c}$ and $\mathcal{E}_t^{c2m}$—based on spatial inclusion, i.e., whether a mesh node falls within a given cell. The positions of the cells $\mathbf{p}_t^c$ are defined as their center coordinates, and the cell features $\bar{\mathbf{v}}_t^c$ are obtained by an average of the connected mesh nodes.

The number of cells—equivalently, the octree level $L$—is a design parameter akin to the world edge radius: mesh nodes falling within the same cell are considered to interact. For large-scale graphs or dense meshes, we split the space with finer resolution cells $C_{L+L_{\text{OCNN}}}$, stored in a $(L + L_{\text{OCNN}})$-level octree, and then apply sparse convolutions to downscale the features to the $L$ level. Details for the sparse convolution refers to section 3.2. We discussed the impact of spatial cells resolution in section A.5.

It's worth noticing that although we do not explicitly model interactions between separate graphs, by aggregating the mesh nodes into spatial cells, it can capture such interactions through cells that encompass nodes from different graphs. Experiments in section 5.3 showed the effectiveness of modeling the interactions by spatial cells.

M2C (mesh-to-cell) message passing is performed to aggregate mesh node features to the corresponding cell features:

$$\bar{\mathbf{v}}_t^c \leftarrow \text{Message-Passing}(\bar{\mathbf{v}}_t^c, \bar{\mathbf{v}}_t^m). \tag{10}$$

Later, to reconstruct features back onto the original graph from the cell representations, a C2M (cell-to-mesh) message-passing operation can be performed:

$$\bar{\mathbf{v}}_t^m \leftarrow \text{Message-Passing}(\bar{\mathbf{v}}_t^m, \bar{\mathbf{v}}_t^c). \tag{11}$$

### 4.2.4 Cell Tokenization

Spatial cells efficiently and effectively capture the structure and information present in the original graph. While OCNN or message-passing operations can be applied to further aggregate the features within each interaction cell, these methods are inherently local—propagating information incrementally through neighborhood connections. This introduces an inductive locality bias into the learned representations. In contrast, attention mechanisms (Vaswani et al., 2017) are designed to overcome such limitations by enabling global feature aggregation in a single step, without relying on local connectivity. This makes them particularly well-suited for capturing long-range dependencies and holistic patterns in complex graphs. For a brief review of our attention module design, see Section A.4.

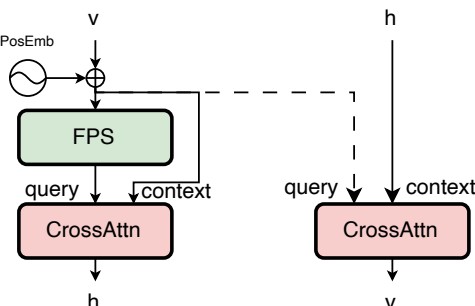

Figure 3: The encoder (left) and decoder (right) cross-attention blocks. We use v to denote feature vectors on the sparse grid (i.e., the previously defined sparse cells), and h to denote the compact latent tokens. Our attention module design are detailed in Section A.4.

We apply cross-attention mechanisms to transform features embedded in spatial cells $C_L$ into compact latent tokens $\mathbf{h}_t$ with a selected dimension $d_{token}$, as illustrated in Figure 3:

$$
\begin{aligned}
\tilde{\mathbf{v}}_t^c &= \mathrm{PosEmb}(\bar{\mathbf{v}}_t^c, \mathbf{p}_t^c), \\
\mathbf{h}_t &= \mathrm{CrossAttn}(\mathrm{FPS}(\tilde{\mathbf{v}}_t^c, \mathbf{p}_t^c), \tilde{\mathbf{v}}_t^c),
\end{aligned}
\tag{12}
$$

where PosEmb and CrossAttn are defined in Section A.4.

### 4.2.5 Processor and Decoder

The latent tokens $\mathbf{h}_t$ are further processed through $L_{SA}$ layers of self-attention modules to condense and integrate global information. During decoding, to reconstruct features on the spatial cells $C_{L,t}$ from the processed latent tokens, we use the positionally embedded features as queries and the latent tokens as context for an cross-attention operation, namely:

$$
\bar{\mathbf{v}}_t^c \leftarrow \mathrm{CrossAttn}(\tilde{\mathbf{v}}_t^c, \mathbf{h}_t).
\tag{13}
$$

The output mesh graph features are first obtained via Equation (11), and the predicted mesh features $\hat{\mathbf{v}}_{t+1}^m$ are then produced by applying an MLP. Similarly, the output element features $\hat{\mathbf{v}}_{t+1}^e$ are computed through an M2E (mesh-to-element) message-passing followed by an MLP.

## 5 Experiments

### 5.1 Experiment Setup

**Datasets** We adopt two representative public datasets from GraphMeshNets (Pfaff et al., 2020) that involve object interactions: **1) DeformingPlate**: A deformable object is compressed by a rigid body, with ~1.3k mesh/4k element nodes per mesh; **2) SphereSimple**: A piece of cloth interacts with a kinematic sphere, with ~2k mesh/4k element nodes per mesh. To further validate our method on large-scale physical simulation tasks—an area where existing literature is limited—we introduce two new datasets: **3) ABCD**: ABCD stands for A Big CAD Deformation, where two deformable objects squish each other, with ~4k mesh/12k element nodes per mesh; **4) ABCD-XL**: follows the same setup as the ABCD dataset, except it uses significantly denser meshes, with ~100k mesh/300k element nodes per mesh.

**ABCD and ABCD-XL** To our knowledge, there were no large-scale physical simulation datasets on a wide variety of geometries available. We constructed a larger and more generalized dataset to fill this gap. The goal of this dataset was to have a wide variety of geometric shapes that are deformed after coming into contact with each other. We used the ABC dataset (Koch et al., 2019), which is a CAD model dataset used for geometric deep learning, to get a wide sample of parts and shapes to deform. Within the ABC dataset, we selected 400 single-part CAD models with relatively higher CAD quality. To generate a simulation, we first

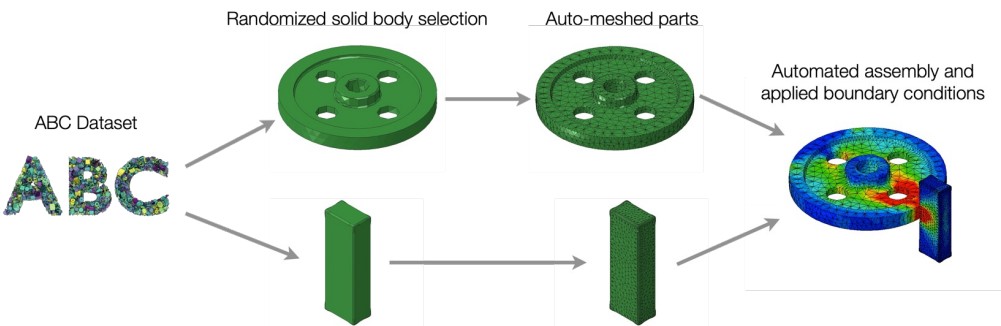

Figure 4: Randomized FEA simulation dataset using geometry from ABC dataset.

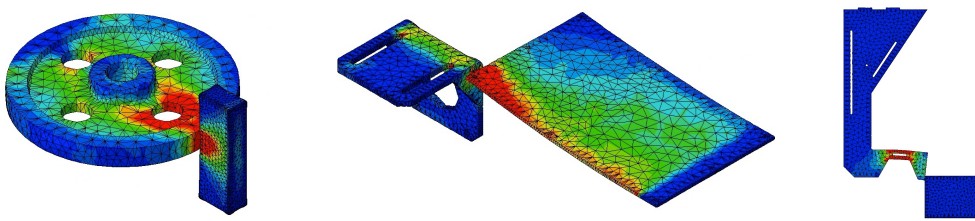

Figure 5: The FEA simulation results using ABC CAD dataset highlight various deformation modes, including compression with associated tension around a hole, as well as plate and beam bending.

randomly select two of these CAD geometries, then auto-mesh them with the meshing tool Shabaka (Hafez & Rashid, 2023). We then align the two meshed parts in 3D space and apply compressive boundary conditions to simulate the parts coming into contact. Figure 4 illustrates the workflow of the dataset construction process. Due to the wide variety of CAD geometries used in the simulations, the resulting deformation exhibit an equally wide variety of deformation modes and stress states within each element. Figure 5 shows several example simulations and the modes of deformation achieved through contact. Using this simulation workflow, we generated two datasets with different mesh resolutions. The ABCD dataset has a mesh size of 4k mesh nodes, and the ABCD-XL dataset has a mesh size that is 25 times larger of 100k mesh nodes.

| Dataset | #Mesh Nodes | Mesh Type | #Steps | System Type | #Samples |
|---|---|---|---|---|---|
| DEFORMINGPLATE | 1271 | Tetrahedral 3D | 400 | Quasi-static | 1000 : 100 : 100 |
| SPHERESIMPLE | 1731 | Triangle 3D | 400 | Newtonian | 1000 : 100 : 100 |
| ABCD | 4408 | Tetrahedral 3D | 20 | Quasi-static | 6000 : 300 : 300 |
| ABCD-XL | 99997 | Tetrahedral 3D | 20 | Quasi-static | 1000 : 100 : 100 |

Table 1: Details of the datasets used in this work. System Type refers to the underlying PDEs. All simulations use linear elements. For DEFORMINGPLATE and SPHERESIMPLE, the dataset providers report using ArcSim (Narain et al., 2012) and COMSOL (De Bézenac et al., 2019), respectively. For ABCD and ABCD-XL, we generated simulations using ABAQUS (Barbero, 2023). #Samples indicates the number of simulations generated for training, validation, and testing, respectively.

**Baselines** We compare our method against several strong baselines across all datasets. **1) MeshGraph-Nets(MGN)**: A state-of-the-art message-passing-based graph neural network; **2) Bi-stride Multi-scale GNN(BSMS)**: Extends MeshGraphNets with bi-stride pooling to construct a U-Net structure for improved scalability; **3) Hierarchical Contact Mesh Transformer(HCMT)**: A Transformer-based architecture specifically designed to model interaction problems using contact-aware mesh transformer blocks. The details of our training settings can be found in Section A.2.

## 5.2 Results

We evaluate all methods on the benchmark datasets by selecting the checkpoints with the lowest validation loss and report their rollout inference accuracy on the test set in Table 2. All experiments on DEFORMINGPLATE, SPHERESIMPLE, and ABCD are conducted on a single machine equipped with 4 V100 GPUs. For the large-scale ABCD-XL dataset, experiments are run on a machine with 8 V100 GPUs. Other training details can be found in Section A.1 and Section A.2. Our method demonstrates superior performance compared to prior methods, achieving a substantial improvement.

| Dataset | | MGN | BSMS | HCMT | Ours |
|---|---|---|---|---|---|
| DEFORMINGPLATE | $\mathbf{u}$ | $5.5 \pm 0.2$ | $5.4 \pm 0.5$ | $2.9 \pm 0.2$ | $\mathbf{1.1 \pm 0.1}$ |
| | $\sigma$ | $6891 \pm 89$ | $10719 \pm 544$ | $7272 \pm 45$ | $\mathbf{4842 \pm 174}$ |
| SPHERESIMPLE | $\mathbf{u}$ | $19.0 \pm 4.9$ | $15.0 \pm 0.8$ | Diverge | $\mathbf{14.4 \pm 0.8}$ |
| ABCD | $\mathbf{u}$ | $0.641 \pm 0.007$ | $0.736 \pm 0.017$ | $0.541 \pm 0.006$ | $\mathbf{0.505 \pm 0.002}$ |
| ABCD-XL | $\mathbf{u}$ | OOM | OOM | OOM | $\mathbf{0.480 \pm 0.002}$ |
| | $\sigma$ | | | | $\mathbf{2.11 \pm 0.82}$ |

Table 2: RMSE (rollout-all, $\times 10^{-3}$ for displacement) evaluation results. $\mathbf{u} = \mathbf{x}_t - \mathbf{x}_0$ denotes displacement and $\sigma$ denotes stress. OOM stands for out-of-memory. RMSE is computed as described in Section A.1, with results shown as mean $\pm$ standard deviation over three independent runs.

## 5.3 Ablation and Parameter Analysis

In this section, we first visualize the spatial cells to demonstrate their effectiveness in capturing interactions, and then present a parameter analysis of key design choices, including the spatial cell length and the number of latent tokens.

**Spatial Cell for Interactions** Existing graph-based methods typically rely on world edges to model interactions. However, computing world edges requires evaluating pairwise distances between mesh nodes, leading to an $O(n^2)$ complexity that limits scalability on large-scale meshes. In contrast, our method leverages spatial quantization to reduce this complexity to $O(n)$ by aggregating nodes into structured cells. We visualize these cells in Figure 6, demonstrating their ability to effectively capture interactions. While it is possible to compute world edges using similar spatial quantization techniques, our approach goes further—by encoding graphs into compact latent tokens, our model combines the expressive power of graph representations with the computational efficiency and global context aggregation capabilities of token-based processing.

**Quantization Cell Length** We run our model on the DEFORMINGPLATE dataset with different $L_{cell}$ values and report the validation loss in Figure 7. When $L_{cell} = 7$, all mesh nodes are assigned to a single cell at the initial frame, tokens then fail to capture interactions among neighboring nodes, degrading accuracy despite the finer resolution. These results verify that grouping a reasonable number of mesh nodes into interaction cells plays a vital role in effectively learning deformable body interactions.

**Number of Latent Tokens** We conduct a parameter search on the number of latent tokens using the DEFORMINGPLATE dataset, reporting both the validation loss and the training epoch time in Figure 8. As shown, too few latent tokens degrade model accuracy, while an excessive number increases training time. Owing to the parallelism of GPUs, model efficiency remains stable up to a certain threshold, beyond which additional tokens improve performance at the cost of runtime. In general, a typical choice for datasets fewer than 10k mesh nodes is 512 latent tokens, which provides sufficient capacity while maintaining comparable training and inference efficiency to smaller values.

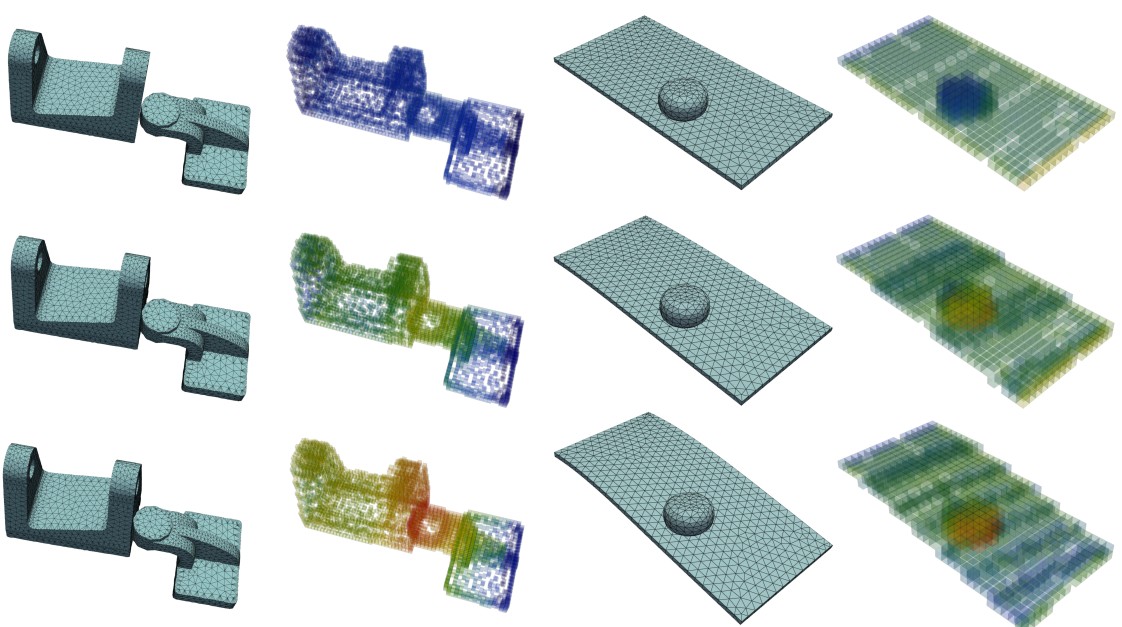

Figure 6: We visualize the spatial cells on the ABCD dataset (left) and DEFORMINGPLATE dataset (right). The figure displays one representative feature channel across the cells. Warmer colors correspond to higher feature norms. Their concentration at the collision regions indicates that the spatial cells effectively capture the critical interactions between the disconnected meshes.

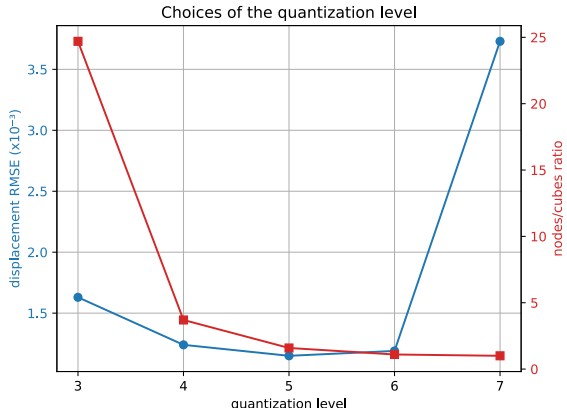

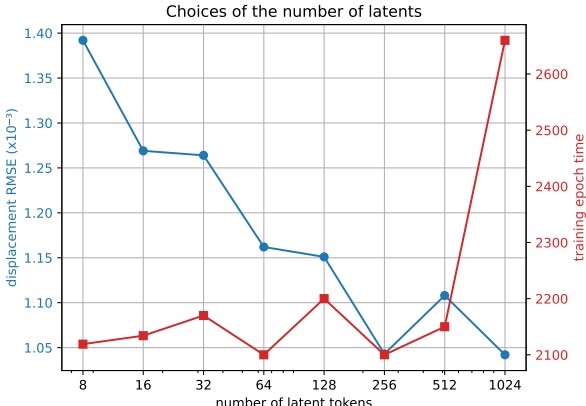

Figure 7: Validation results on the DEFORMING-PLATE dataset with different $L_{cell}$ settings. The nodes/cells ratio denotes the average number of mesh nodes per cell at the initial frame.

Figure 8: Validation loss and training epoch time on the DEFORMINGPLATE dataset for varying numbers of latent tokens. Accuracy generally improves with more latent tokens, while training time remains stable until increasing sharply beyond a threshold.

## 5.4 Computational Efficiency and Scaling Capability

The training and inference times are reported in Table 3. All experiments are conducted on a machine equipped with 4 V100 GPUs. While increasing the batch size significantly improves training efficiency (e.g., 2.8× for MGN, 4.8× for BSMS, and 3.1× for ours when using a batch size of 48 on SPHERESIMPLE), we ensure a fair comparison by fixing the batch size to 4 (i.e., 1 per GPU) across all methods. Our method

demonstrates comparable training and inference efficiency to state-of-the-art graph-based approaches on small-scale mesh size.

| Model | DEFORMINGPLATE | | SPHERESIMPLE | | ABCD | | ABCD-XL | |
|---|---|---|---|---|---|---|---|---|
| | Train | Val | Train | Val | Train | Val | Train | Val |
| MGN | 9393 | 316 | 8744 | 361 | 3425 | 215 | - | - |
| BSMS | 17700 | 410 | 12754 | 508 | 5442 | 332 | - | - |
| HCMT | 16450 | 470 | 12913 | 510 | 5862 | 328 | - | - |
| Ours | 9333 | 392 | 7282 | 394 | 2742 | 188 | 16435 | 8123 |

Table 3: Training and inference epoch time (seconds) evaluation. The reported epoch time refers to the total time taken for a single pass over the entire dataset during training or inference. For a fair comparison, all models are evaluated on a 4-GPU node with a total batch size of 4.

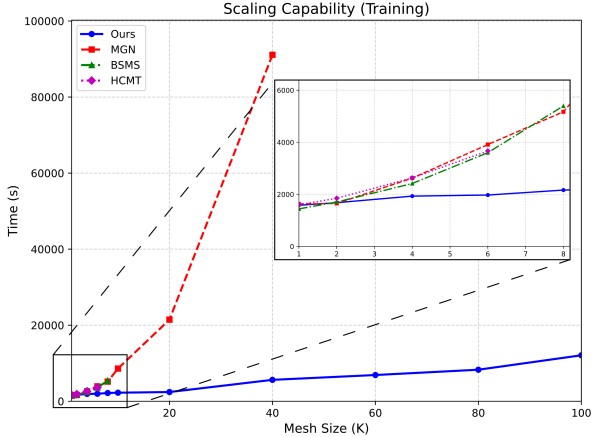

Figure 9: Training time per epoch across different mesh sizes. MGN, BSMS, and HCMT run out of memory at mesh sizes beyond 40K, 8K, and 6K, respectively.

Figure 10: Inference time per epoch across different mesh sizes.

To further evaluate scalability, we conducted experiments on the ABCD-XL dataset by generating subgraphs with varying mesh sizes. We compared the training and inference time of different methods on a machine equipped with 4 V100 GPUs, and we used the training set to evaluate both training and inference efficiency. As shown in Figure 9-10, all methods exhibit similar performance in the small-scale regime. However, our method demonstrates significantly better scalability as the element size exceeds 20k.

Although the BSMS method demonstrated good scalability on surface meshes in its original paper, we observed that it scales poorly on volume meshes due to the increased number of bi-stride edges introduced during pooling. On surface meshes, bi-stride edges consistently downsample upper-layer edges. However, in the case of volume meshes, the edge count can grow significantly. For example, in a volume mesh graph with 21k mesh nodes, the number of nodes and edges across a 6-layer BSMS model are: 21k/210k, 11k/276k, 5.7k/534k, 3.2k/1.8M, 1.8k/1.6M, 0.9k/483k.

## 5.5 Visualizations

We present a prediction result on the ABCD dataset in Figure 11, and DEFORMINGPLATE dataset on Figure 12. Further visualizations are provided in Section A.3.

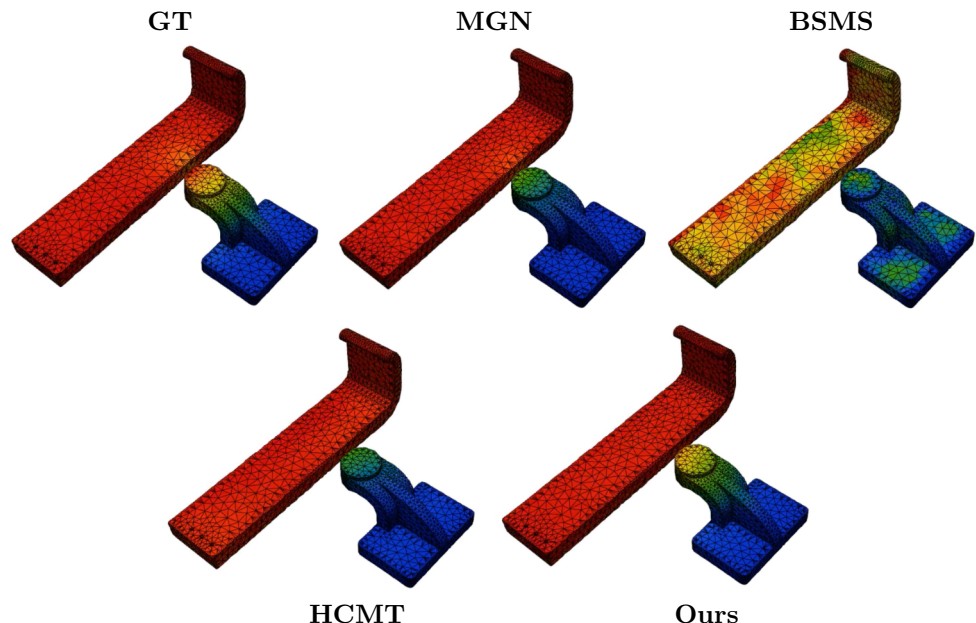

Figure 11: Visualization results on the ABCD dataset. Displacement is visualized using color warmth, with warmer tones indicating greater displacement magnitude. Our method yields predictions that are closer to the ground truth at the collision region.

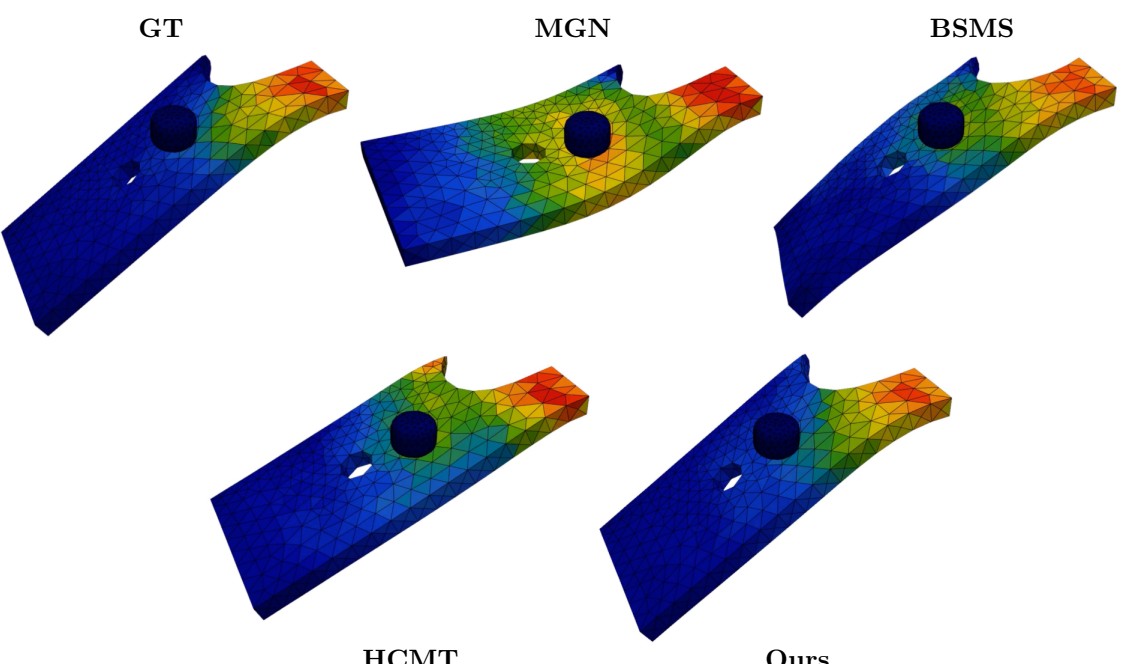

Figure 12: Visualization results on the DEFORMINGPLATE dataset. Stress is visualized using color warmth, with warmer tones indicating greater stress magnitude. Our method produces shapes that more closely match the ground truth, indicating more accurate displacement predictions, while the red-colored cells highlight correspondingly closer stress predictions.

## 6    Limitation and Future Work

Despite its effectiveness, the method has certain limitations. Specifically, the octree structure enforces a fixed set of cell lengths across levels, making it difficult to identify an optimal configuration. As shown in Figure 7, the performance curve remains relatively flat near the optimal $L_{cell}$, but a more flexible and finer grid control could yield further improvements. Besides, we employed FPS and CA to construct an efficiency-oriented bottleneck, yet recent work such as AnchorFormer Shan et al. (2025) introduces anchor-based approximations for full attention, offering a promising alternative that can preserve efficiency while potentially improving accuracy. Furthermore, adding additional constraints on physical laws such as the conservation of energy, enforcing boundary conditions, or satisfying the underlying PDEs could also further improve the models performance and generalization.

## 7    Conclusion

In this paper, we proposed a new method that encodes graphs into compact tokens by leveraging sparse 3D operations, followed by transformer-based processing for expressive learning. This strategy combines the structural richness of graphs with the scalability and efficiency of 3D computation, enabling our model to scale to large inputs without compromising accuracy.

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

# A  Technical Appendices and Supplementary Material

## A.1  Dataset Details

**Dataset Settings**  In Table 4 we list details of all the dataset used in the experiments.

| Dataset | System | Solver | Mesh type | steps | $\Delta t$ | $r_W$ |
|---------|--------|--------|-----------|-------|-----------|-------|
| SPHERESIMPLE | cloth | ArcSim | triangle 3D | 500 | 0.01 | 0.05 |
| DEFORMINGPLATE | hyper-el. | COMSOL | textrahedral 3D | 400 | - | 0.03 |
| ABCD/ABCD-XL | hyper-el. | Abaqus | textrahedral 3D | 21 | - | 0.003 |

Table 4: Dataset details. $r_W$ denotes the world edge radius, meaning that nodes within a distance of $r_W$ are considered connected by world edges. Note that world edges are only computed for MGN, BSMS, and HCMT.

Our method uses slightly different input features with Pfaff et al. (2020), as we do not explicitly compute world edges. The input and output features used in our method are summarized in Table 5. Specifically, $\mathbf{v}_i^m$ and $\mathbf{v}_i^e$ represent the input features on mesh and element nodes, respectively, while $\mathbf{v}_o^m$ and $\mathbf{v}_o^e$ denote the corresponding output features. $\mathbf{e}_i^{m2m}$ and $\mathbf{e}_i^{e2m}$ are input edge features for mesh-to-mesh (m2m) and element-to-mesh (e2m) edges.

We use $\mathbf{n}$ and $\mathbf{m}$ to denote node type and material type, respectively. The node type indicates whether a mesh node is a boundary node—such nodes have predefined next-frame values and therefore do not require updates. $\sigma$ denotes stress.

For MGN, BSMS, and HCMT, the input and output features follow the exact definitions from Pfaff et al. (2020) on SPHERESIMPLE and DEFORMINGPLATE. We note that in the BSMS paper, the authors included velocity as an input feature for DEFORMINGPLATE. In contrast, we follow the original MeshGraphNets (MGN) implementation, which uses node type and (relative) positions as input. On ABCD, the definitions of $\mathbf{v}_i^m$, $\mathbf{e}_i^{m2m}$, and $\mathbf{v}_o^m$ are consistent with that in Table 5, and the world edges are associated with edge features $\mathbf{p}_t^{ij} \big\| \|\mathbf{p}_t^{ij}\|$.

| Dataset | $\mathbf{v}_i^m$ | $\mathbf{v}_i^e$ | $\mathbf{e}_i^{m2m}$ , $\mathbf{e}_i^{e2m}$ | $\mathbf{v}_o^m$ | $\mathbf{v}_o^e$ | history |
|---------|------------------|------------------|---------------------------------------------|------------------|------------------|---------|
| SPHERESIMPLE | $\mathbf{n}, \dot{\mathbf{p}}_t$ | $\dot{\mathbf{p}}_t$ | $\mathbf{p}_0^{ij}\big\|\|\mathbf{p}_0^{ij}\|\big\|\mathbf{p}_t^{ij}\big\|\|\mathbf{p}_t^{ij}\|$ | $\ddot{\mathbf{p}}_t$ | - | 1 |
| DEFORMPLATE | $\mathbf{n}, \mathbf{u}_t$ | $\mathbf{u}_t$ | $\mathbf{p}_0^{ij}\big\|\|\mathbf{p}_0^{ij}\|\big\|\mathbf{p}_t^{ij}\big\|\|\mathbf{p}_t^{ij}\|$ | $\dot{\mathbf{p}}_t, \sigma_{t+1}$ | - | 0 |
| ABCD | $\mathbf{n}, \mathbf{m}, \mathbf{u}_t$ | $\mathbf{u}_t$ | $\mathbf{p}_0^{ij}\big\|\|\mathbf{p}_0^{ij}\|\big\|\mathbf{p}_t^{ij}\big\|\|\mathbf{p}_t^{ij}\|$ | $\dot{\mathbf{p}}_t$ | - | 0 |
| ABCD-XL | $\mathbf{n}, \mathbf{u}_t$ | $\mathbf{m}, \sigma_t, \mathbf{u}_t$ | $\mathbf{p}_0^{ij}\big\|\|\mathbf{p}_0^{ij}\|\big\|\mathbf{p}_t^{ij}\big\|\|\mathbf{p}_t^{ij}\|$ | $\dot{\mathbf{p}}_t$ | $\sigma_{t+1}$ | 0 |

Table 5: Input and output features for our method.

**World Edge Calculation**  We adopt the world-edge construction method proposed in the MGN paper, with modifications to accommodate the dense meshes in the ABCD dataset. Compared to DEFORMINGPLATE and SPHERESIMPLE, the meshes in ABCD are significantly denser, which causes the original world-edge computation to sometimes produce world edges even larger than mesh edges. To address this, we retain only the 1000 world edges with the smallest pairwise distances.

**RMSE Calculation**  The Root Mean Square Error (RMSE) is computed in a per-sequence manner: we first calculate the mean squared error for each sequence, then take the square root, and finally average the RMSE across all sequences in the dataset.

## A.2 Training Settings

For all datasets, we adopt a pairwise training strategy where a graph is randomly selected from a sequence as the input, and its subsequent graph is used as the target. We follow the same training noise strategy as proposed in Pfaff et al. (2020). The noise scale is set to 0.003 for both the ABCD and ABCD-XL datasets. The input and target features are all normalized to zero mean and unit variance based on the statistics of the training set. All experiments are conducted on a machine equipped with four V100-32GB GPUs, unless otherwise specified.

We did some modifications to the training process for a improved performance:

**Batch Size** We increased the training batch size from 1 or 2 to 48 (12 per GPU on a 4-GPU node) for MGN and BSMS which showed a much faster training procedure. HCMT inherently not applicable on batched graphs, so we kept a batch size with 1 per GPU.

**Learning Rate** We adopt square root scaling for the learning rate with respect to batch size. Starting with a base learning rate of 0.0001 for a batch size of 2, the final learning rate $LR$ for a batch size of 48 is computed as $0.0001 \times \sqrt{48/2} \approx 0.00049$. The learning rate is linearly warmed up from $0.0001LR$ to $LR$ over the first 2000 steps, followed by cosine decay to zero at the 101st epoch (training stops at the 100th epoch).

For purely graph-based methods—namely MGN, HCMT, and BSMS—we found that the learning rate scheduling strategy facilitates faster convergence, while the square root scaling strategy had a negative effect. Therefore, we retain the scheduling strategy and use a fixed learning rate of 0.0001 across all datasets.

**Training Iterations** We extend the training iterations from 5M steps (approximately 25 epochs) to 100 epochs.

**Loss** We use mean squared error (MSE) loss across all experiments. For the DEFORMINGPLATE and ABCD-XL datasets, we adopt multi-head outputs to jointly predict displacement and stress, assigning loss weights of 1 and 0.01, respectively.

## A.3 Further Discussion

**MGN** MeshGraphNets (MGN) is a strong baseline for mesh-based physical simulations due to the expressiveness of its graph-based representation. However, this expressiveness also introduces several challenges:

- **Edge overhead:** The computational burden in graph models often arises from the edge set, which can be several times larger than the node set. This issue is exacerbated on large-scale meshes, where edge-based feature aggregation results in significant computational and memory overhead.

- **Limited global context:** Message-passing operations in graphs are inherently local, requiring many iterations to propagate information across distant nodes. For meshes with over 100K nodes, hundreds of message-passing steps may be needed to fully capture global interactions.

- **Scalability of world edges:** On dense meshes, the number of world edges can grow prohibitively large. This not only increases computation but also makes it difficult to distinguish meaningful interactions from spurious ones.

**BSMS** BSMS introduces the bi-stride pooling mechanism to address some of the limitations of MGN. By recursively down-scaling the graph—halving the mesh size at each stage—the method reduces both the number of nodes and edges, allowing for faster propagation of global information. While this strategy proves effective for small to medium-scale meshes, it still inherits the structural limitations of graph-based methods when applied to large-scale problems. In industrial FEA simulations where mesh sizes can exceed 100K elements, the graph structure remains a bottleneck. Moreover, as discussed in Section 5.4, the bi-stride pooling algorithm fails to generalize effectively to volume meshes, limiting its applicability to 3D deformable body problems. Furthermore, although pooling edges can be precomputed—thereby accelerating training—the precomputation

process involves matrix multiplications whose complexity scales with the number of mesh nodes. This becomes prohibitively expensive for large meshes, limiting the applicability of BSMS in high-resolution simulation settings.

**HCMT**   HCMT incorporates attention mechanisms to address one of the key limitations of MGN—its inefficiency in aggregating global information. However, the attention computations in HCMT are performed directly on each mesh node. While they modify the original attention formulation to avoid the $O(n^2)$ complexity associated with standard attention matrices, this comes at the cost of reduced theoretical soundness. Moreover, despite these modifications, the method still scales poorly to large-scale meshes, limiting its practicality in high-resolution simulation tasks.

On the SPHERESIMPLE dataset, HCMT performs well during the initial 50 frames but gradually diverges thereafter. This suggests that, while HCMT is effective on the quasi-static DEFORMINGPLATE dataset, it may face difficulty generalizing to dynamic problems like SPHERESIMPLE.

### A.4   Transformer Operations

Following the design in Zhang et al. (2023a), originally developed for point clouds, we extend this framework to handle vector fields.

**Positional Embedding**   Given a feature set $c$ with associated positions $p^c$, we first inject spatial information into the features via positional embedding:

$$c' = c + \text{PosEmb}(p^c), \tag{14}$$

where $\text{PosEmb} : \mathbb{R}^d \to \mathbb{R}^{d_c}$ is a column-wise embedding function that maps input positions $p^c$ (with $d \in \{2, 3\}$) to the feature space of dimension $d_c$, matching the dimensionality of $c$.

**Attention**   An attention operation is defined to aggregate three feature vectors $q \in \mathbb{R}^{N_q \times d_q}$, $k \in \mathbb{R}^{N_k \times d_k}$, $v \in \mathbb{R}^{N_v \times d_v}$, where $N_k, N_q, N_v \in \mathbb{R}$ are sequence lengths and $d_q, d_k, d_v \in \mathbb{R}$ are feature dimensions,

$$\text{Attention}(q, k, v) = \text{softmax}(\frac{qk^T}{\sqrt{d_k}})v. \tag{15}$$

A Multi-head Attention (MHA) operation is defined as

$$\begin{aligned}
\text{MHA}(q, k, v) &= \text{Concat}(\text{head}_1, ..., \text{head}_h)W^o, \\
\text{head}_i &= \text{Attention}(qW_i^q, kW_i^k, vW_i^v).
\end{aligned} \tag{16}$$

**Transformer Blocks**   Building on the multi-head attention mechanism and adopting a pre-norm structure, we construct two types of Transformer blocks, as illustrated in Figure 13.

- CrossAttn: Given a set of query features $c_{\text{query}}$ and context features $c$context, the Cross-Attention operation

$$h = \text{CrossAttn}(c_{\text{query}}, c_{\text{context}}) \tag{17}$$

  aggregates information from $c$context into $c_{\text{query}}$. The output $h$ maintains the same length as the query, making this block particularly useful for compressing or decompressing representations of the context features.

- SelfAttn: This is the standard self-attention mechanism where the query and context features are identical, i.e.,

$$h = \text{SelfAttn}(c). \tag{18}$$

FFN stands for Feed-Forward Network. In all our CrossAttn and SelfAttn blocks, we use an FFN module with GEGLU activation as described in Shazeer (2020).

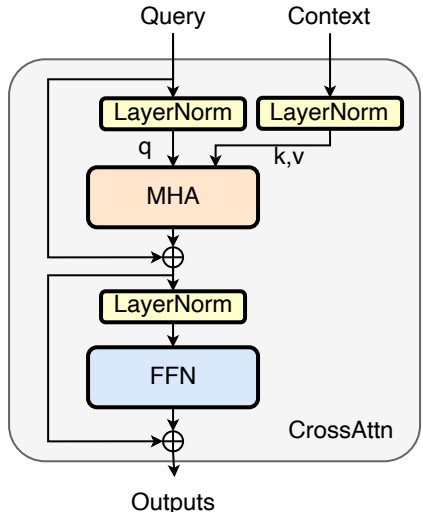 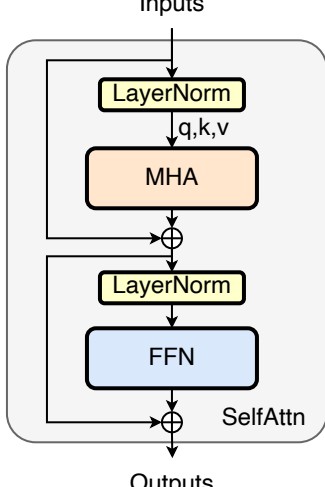

Figure 13: Transformer blocks. The CrossAttn is displayed on the left, while the SelfAttn is displayed on the right.

## A.5 Model Details

**MLP**    The MLPs used in input, message-passing and output layers are two-layer MLPs with ReLU activations with output size of 128. The hidden layer size of message-passing and input MLPs are 128, while the output MLPs are 32. The outputs of the message-passing MLPs are further normalized using LayerNorm. To enhance their effectiveness, we adopt Random Weight Factorization (RWF, (Wang et al., 2022)).

**Transformer**    We use attention layers with 8 heads, each with a feature dimension of 64. The query, key, and value projections are implemented using bias-free linear layers, while the output projection includes a bias term. We adopt a pre-norm setup, applying LayerNorm before both the attention and FFN layers. The FFN follows the GEGLU formulation as described in Shazeer (2020), and has a hidden dimension of 512. Dropout with a rate of 0.1 is applied within both the attention and FFN modules.

**Model Architecture**    The input and output features on graph nodes and edges are normalized using statistics computed from 400 randomly selected graph pairs from the dataset. Our model begins with an input encoding MLP that maps node and edge features into a hidden space of size 128. This is followed by a one-step message-passing operation—either m2m or e2m—to aggregate information onto the mesh nodes.

| Dataset | $L_{cell}$ | $l_{ocnn}$ | $d_{token}$ | $L_{SA}$ |
|---------|-----------|-----------|------------|----------|
| SPHERESIMPLE | 5 | 0 | 256 | 12 |
| DEFORMINGPLATE | 5 | 0 | 256 | 12 |
| ABCD | 8 | 0 | 512 | 12 |
| ABCD-XL | 12 | 4 | 512 | 12 |

Figure 14: Model hyperparameter.

Mesh node positions are then quantized using an $L_{cell}$-layer octree, where $L_{cell}$ is a hyperparameter. A one-step message-passing is used to aggregate mesh node features into each cell. Optionally, an OCNN module with $l_{ocnn}$ layers (also a hyperparameter) is applied to downscale the features from the $L_{cell}$-th octree layer to the $(L_{cell} - l_{ocnn})$-th layer. Each OCNN layer consists of a sparse convolution with a $3 \times 3 \times 3$ kernel and a stride of 2, which effectively moves features up one level in the octree hierarchy.

To obtain a compact latent representation, a cross-attention layer is applied to map features on the sparse cells to a fixed set of tokens of dimension $d_{token}$. These tokens are then processed using $L_{SA}$ self-attention transformer blocks.

The decoding process mirrors the encoder. We first apply a cross-attention mechanism to decode the latent tokens back to the octree features. These features are then upscaled to the original $L_{cell}$-layer resolution using a transposed sparse convolution-based OCNN with the same number of layers as in the encoder. A one-step message-passing operation is performed to decode the cell features back to the mesh nodes. The resulting features are concatenated with the original mesh node features from the input encoding MLP to produce $\mathbf{v}'^m_t$, which is then passed through an output MLP to generate the final predictions. If output features are also required on element nodes, an additional one-step message-passing—without edge features—is performed from mesh to element nodes, followed by a separate output MLP.

The fore-mentioned hyperparameter for each dataset are listed in Table 14.

### A.6  Farthest Point Sampling Implementation

We outline the FPS algorithm in Algorithm 1, while noting that readers may directly use existing open-source implementations.

---

**Algorithm 1** Farthest Point Sampling (FPS)

---

**Require:** Point set $P = \{p_1, \ldots, p_N\}$, sample size $K$
**Ensure:** Sample indices $S$ of size $K$
 1: Initialize $S \leftarrow \{\}$.
 2: Randomly choose initial index $i_0$.
 3: $S \leftarrow S \cup \{i_0\}$.
 4: **for** each $i \in \{1, \ldots, N\} \setminus S$ **do**
 5:     $d[i] \leftarrow \text{dist}(p_i, p_{i_0})$
 6: **end for**
 7: **for** $t \leftarrow 2$ to $K$ **do**
 8:     $j \leftarrow \arg\max\limits_{i \notin S} d[i]$
 9:     $S \leftarrow S \cup \{j\}$
10:     **for** each $i \notin S$ **do**
11:         $d[i] \leftarrow \min\big(d[i], \ \text{dist}(p_i, p_j)\big)$
12:     **end for**
13: **end for**
14: **return** $S$

---

## B  Additional Experiments

### B.1  MAPE Evaluation

For completeness, we also report the Mean Absolute Percentage Error (MAPE) on the DEFORMINGPLATE, SPHERESIMPLE, and ABCD datasets in Table 6. MAPE measures the relative prediction error as a percentage and is scale-independent, but it is known to be unstable when ground-truth values are close to zero. This issue is particularly relevant for DEFORMINGPLATE and ABCD, where simulated objects are centered around zero. To mitigate, we mask out ground-truth values with $\|y_t\|_2 < 10^{-5}$. The MAPE used in this paper is defined as:

$$\frac{1}{T} \sum_{t=1}^{T} \text{MASK}\left( \frac{\|\hat{y}_t - y_t\|_2}{\|y_t\|_2} \right), \tag{19}$$

where $y_t$ and $\hat{y}_t$ denote the ground-truth and predicted values at time $t$, and $\text{MASK}(\cdot)$ discards terms with $\|y_t\|_2 < 10^{-5}$. We emphasize that these results are provided only as a reference; RMSE remains the primary and more reliable metric for comparison across methods.

| Dataset | | | MGN | BSMS | HCMT | Ours |
|---|---|---|---|---|---|---|
| DEFORMINGPLATE | $\mathbf{u}$ | | $0.293 \pm 0.006$ | $2.099 \pm 0.0175$ | $0.391 \pm 0.023$ | $\mathbf{0.132 \pm 0.008}$ |
| SPHERESIMPLE | $\mathbf{u}$ | | $0.059 \pm 0.001$ | $0.031 \pm 0.001$ | Diverge | $\mathbf{0.030 \pm 0.001}$ |
| ABCD | $\mathbf{u}$ | | $\mathbf{0.626 \pm 0.012}$ | $1.732 \pm 0.033$ | $0.673 \pm 0.001$ | $0.691 \pm 0.004$ |

Table 6: The Mean Absolute Percentage Error (MAPE) (rollout-all) evaluation results.

## B.2 Rollout Visulization

We select one representative case from DEFORMINGPLATE, SPHERESIMPLE, and ABCD, and show their visualizations in Figures 15–17. The per-frame RMSE of the selected cases, along with the average over the test datasets, is reported in Figures 18–20.

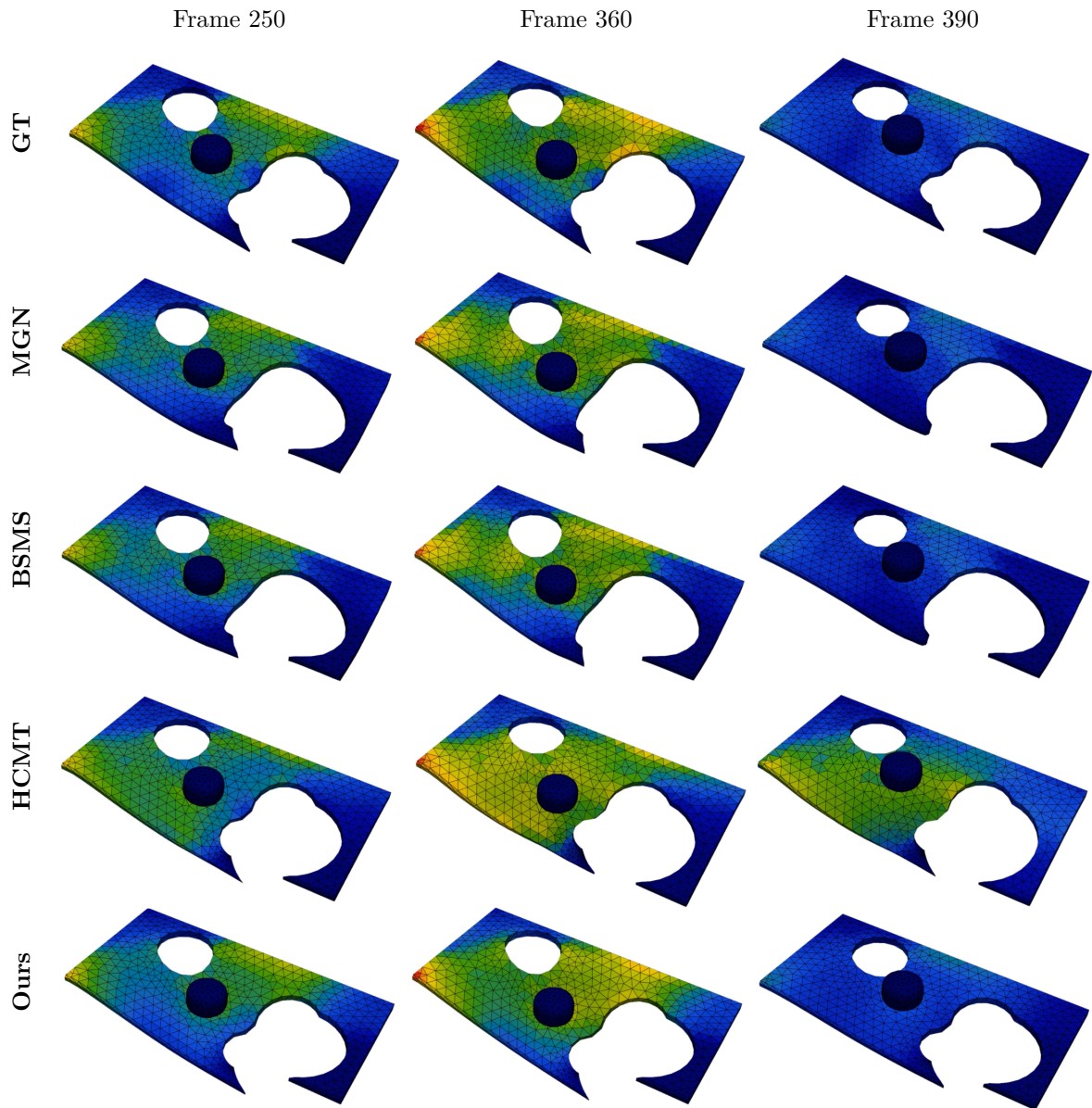

Figure 15: Rollout visualization results on the DEFORMINGPLATE dataset.

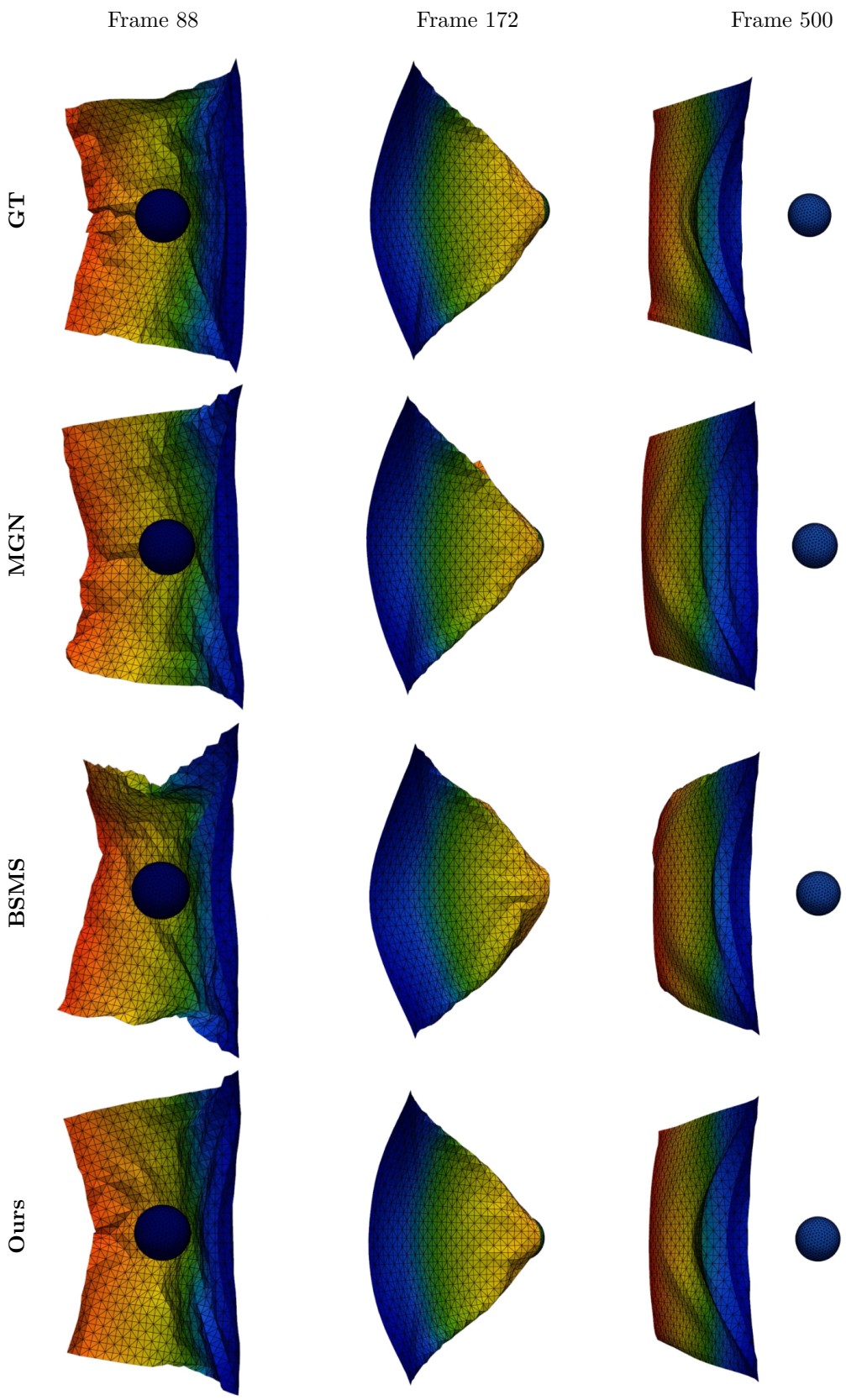

Figure 16: Rollout visualization results on the SPHERESIMPLE dataset.

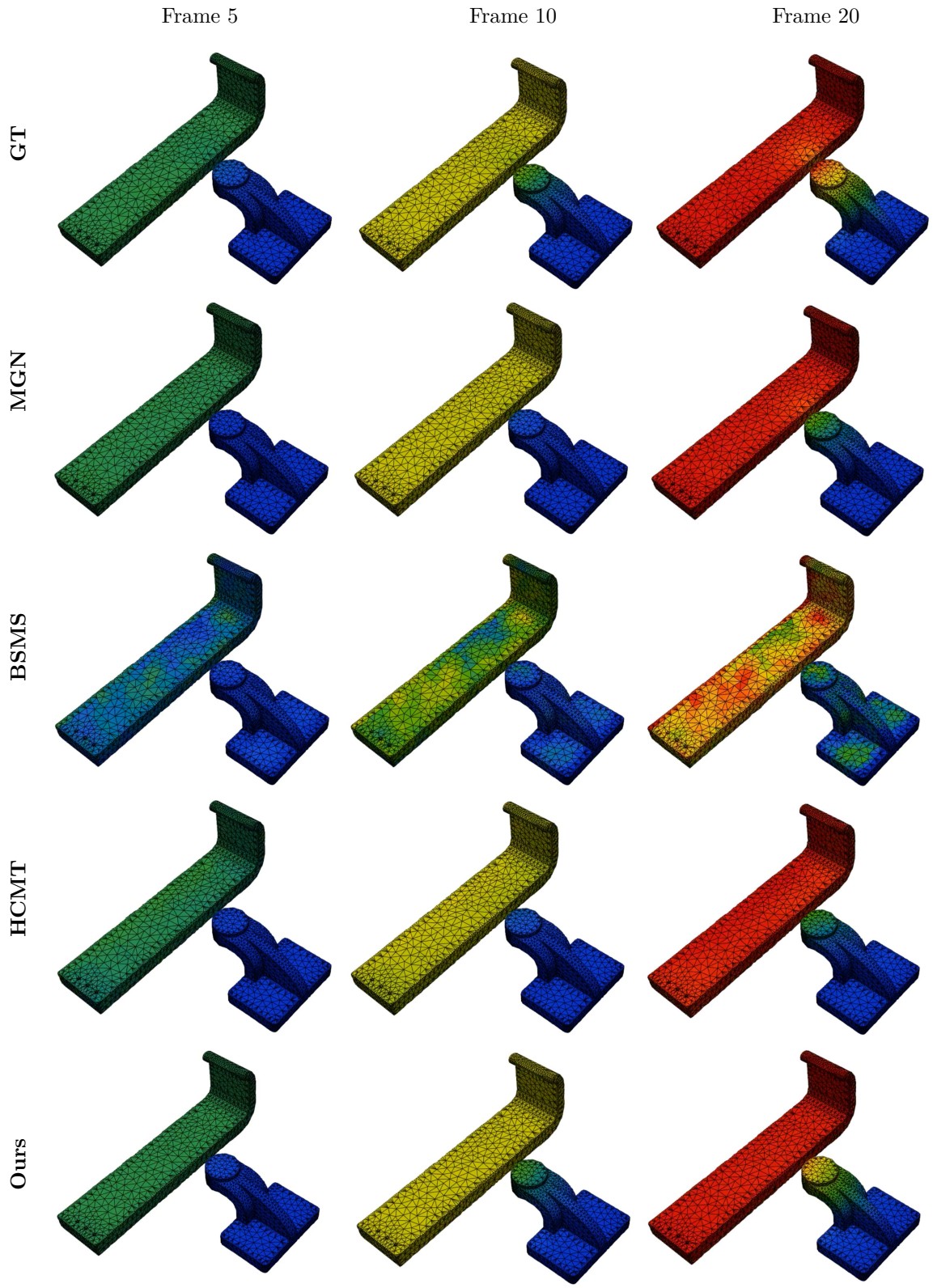

Figure 17: Rollout visualization results on the ABCD dataset.

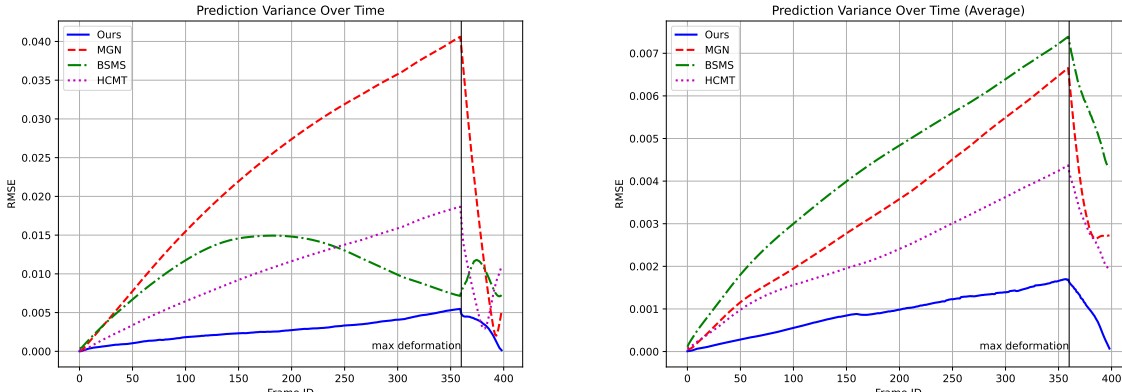

Figure 18: Prediction variance over time on the DEFORMINGPLATE dataset. The deformation increases and reaches its maximum at the 360th frame across all cases. Our method exhibits a smaller growth rate with respect to deformation and successfully recovers to the initial state once the deformation ends.

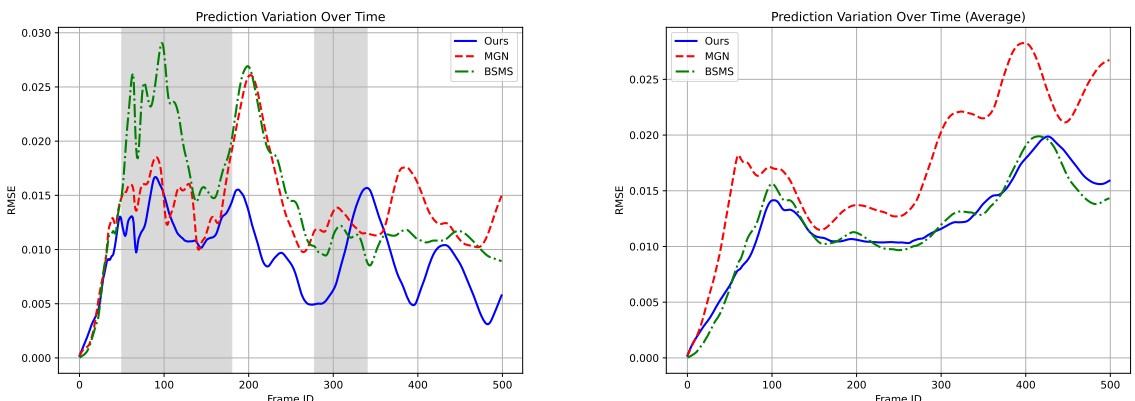

Figure 19: Prediction variance over time on the SPHERESIMPLE dataset. The gray region marks the interaction period between the ball and the cloth. After separation, our method maintains consistent predictions.

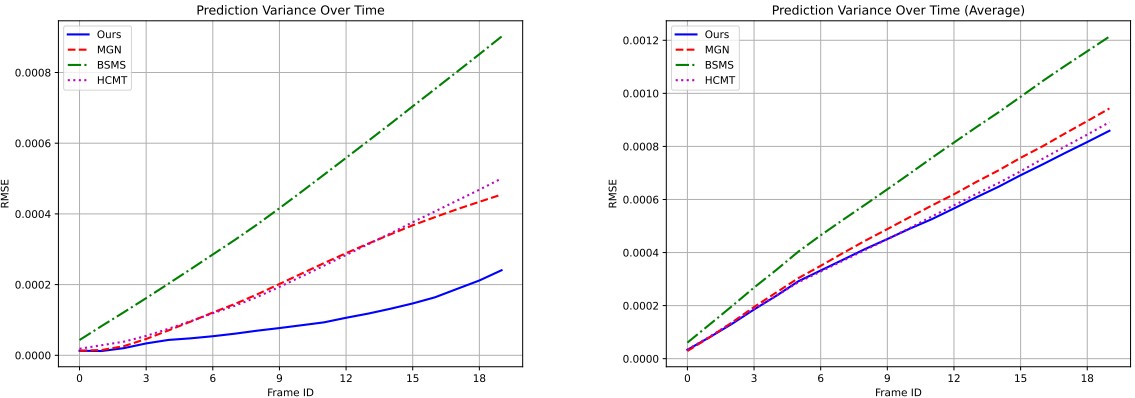

Figure 20: Prediction variance over time on the ABCD dataset. The maximum deformation occurs at the final frame.

