# OpenReview forum: "Learning Deformable Body Interactions With Adaptive Spatial Tokenization"
_TMLR — Accepted by TMLR_

### Review · Reviewer_Zaye · 2025-09-07

**Summary Of Contributions:**

This paper puts forward a novel method named Adaptive Spatial Tokenization (AST) for effectively simulating the interactions among deformable bodies. The main contributions of the paper include:

1. **Innovative Spatial Tokenization Method**: It presents the AST method. This method naturally clusters adjacent mesh nodes by partitioning the simulation space into grid cells and projecting the unstructured mesh onto a structured grid.
2. **Attention-Based Model**: The authors develop an attention model that functions on spatial tokens within a latent space to simulate the interactions of deformable bodies.
3. **Large-Scale Dataset**: The paper introduces the ABCD-XL dataset, which encompasses diverse scenarios of deformable body interactions.

**Main Strengths:**

- **Scalability Strength**: The method can handle large-scale meshes with more than 100,000 nodes. In contrast, existing methods encounter failures because of memory limitations.
- **Computational Efficiency**: It reduces the complexity from O(n²) to O(n) by avoiding the costly construction of pairwise global edges.
- **Adequate Experimental Validation**: Comprehensive experimental validations and ablation studies have been carried out on four datasets.
- **Larger and More Generalized Dataset**: The ABCD-XL dataset appears to be challenging and has the potential to drive the development of this field.

**Main Weaknesses:**

- **Treatment of Physical Constraints**: There is an inadequate discussion regarding how to maintain physical conservation laws, such as the conservation of energy and the conservation of momentum.
- **Lack of Introduction to the ABCD-XL Dataset**: Considering that the dataset is regarded as a contribution by the authors, the paper lacks an introduction to the specific statistical characteristics of this dataset and a comparison with other datasets.

**Additional Comments:**

N/A

**Audience:**

Yes

**Audience Explanation:**

The research findings of this paper have significant value and broad interest for the TMLR audience:

- **Practical Application Value**: It has direct application potential in areas including materials science, mechanical design, and robotics.
- **Computational Efficiency Improvement**: This paper offers a novel solution for industrial-grade simulations, which is of great practical importance.
- **Large-Scale Simulation Capability**: It bridges the gap in existing methods when dealing with complex, large-scale deformable body interactions.

**Claims And Evidence:**

Yes

**Claims Explanation:**

The main claims of the paper are supported by accurate, convincing, and clear evidence:

- Comprehensive quantitative evaluation on four datasets (DeformingPlate, SphereSimple, ABCD, ABCD-XL).
- Fair comparison with three strong baseline methods (MGN, BSMS, HCMT).
- Provided detailed RMSE metrics and visualization results.

**Requested Changes:**

- Provide a more detailed introduction to the proposed ABCD-XL dataset, including its advantages over other datasets.
- Lack of stability analysis of the method under extreme conditions (e.g., large deformation, contact separation).
- Add an analysis of how physical constraints such as conservation of energy and conservation of momentum are maintained in the AST framework.
- Conduct a sensitivity analysis of crucial hyperparameters, such as Lcell and dtoken. This analysis will help in understanding how variations in these hyperparameters impact the performance of the proposed method.
- Add a comparison of accuracy and efficiency with commercial FEM software on the same problem.
- Explore and discuss the scenarios and limiting conditions under which the method might encounter failures.
- Commit to making the code and the ABCD-XL dataset publicly available upon acceptance. Open-sourcing the code and dataset will significantly enhance the reproducibility of the research.

---

> ### Author Response · Authors · 2025-09-25
>
> We are grateful for the time and effort you dedicated to carefully reviewing our manuscript and offering constructive suggestions.
>
> **1. Provide a more detailed introduction to the proposed ABCD-XL dataset, including its advantages over other datasets.**
>
> Thank you for this suggestion. We have revised the manuscript to provide a clearer introduction to the ABCD and ABCD-XL datasets:
> * We moved the dataset introduction from the Appendix into the main paper (Section 5.1).
> * We added a comparative dataset table (Table 1) to highlight differences between the four datasets used in our experiments: DeformingPlate, SphereSimple, ABCD, and ABCD-XL.
> * We expanded the description of ABCD and ABCD-XL to emphasize their novelty, scale, and diversity.
>
> Comparing with other dataset, ABCD-XL has advantages for following aspects:
> * The ABCD datasets are built on the large-scale CAD collection from the ABC dataset \citep{koch2019abc}, ensuring a wide variety of shapes and geometries not covered in prior simulation datasets.Each simulation involves two randomly selected CAD parts undergoing compressive contact, producing diverse deformation modes (compression, bending, tension, stress concentration around features).
> * The ABCD dataset provides medium-resolution meshes (about 4k nodes), while ABCD-XL pushes the scale significantly further (about 100k nodes, 25× larger). To the best of our knowledge, ABCD-XL is the first large-scale physical simulation dataset with six-figure mesh node counts, enabling benchmarking of machine learning methods in high-resolution, computationally challenging regimes.
> * Unlike existing datasets (e.g., DeformingPlate, SphereSimple), ABCD-XL captures both geometric diversity and simulation scale, allowing models to be evaluated on their ability to generalize across shapes while handling large-scale PDE systems.
>
> **2. Lack of stability analysis of the method under extreme conditions (e.g., large deformation, contact separation).**
>
> We have added per-frame RMSE plots in Figures 18–20 to analyze the stability of our method under extreme conditions. On the DeformingPlate dataset, the deformation increases and reaches its maximum at the 360th frame, while on the ABCD dataset, it grows continuously and reaches its maximum at the final frame. In both cases, our method exhibits a smaller growth rate with respect to deformation, resulting in lower prediction errors for large deformations, and successfully recovers to the initial state once the deformation ends. On the SphereSimple dataset, where the ball and cloth interact and later separate, our method maintains consistent predictions across both interaction and separation periods.
>
> **3. Add an analysis of how physical constraints such as conservation of energy and conservation of momentum are maintained in the AST framework.**
>
> We did not explicitly add any physical constraint to maintain conservation of energy, but this is a good suggestion which we’ve added to our future work section. We do have an implicit constraint on conservation of momentum for the SphereSimple dataset because this is the only dataset which is in the dynamic regime. The other datasets are quasi-static and have velocity and momentum of 0 everywhere. To conserve momentum, we make our predictions on the acceleration of each node, then update the previous time step’s velocity using a forward difference scheme. This creates an implicit bias that enables the model to only increment on the previous time step’s velocity, and consequently preserve the previous time step’s momentum. However, this is not novel to our work, and the original MGN paper discusses this same approach.
>
> **4. Conduct a sensitivity analysis of crucial hyperparameters, such as Lcell and dtoken. This analysis will help in understanding how variations in these hyperparameters impact the performance of the proposed method.**
>
> We have added an analysis of Lcell and dtoken in Section 5.3, under the paragraphs Quantization Cell Length and Number of Latent Tokens. In general, varying the quantization level (Lcell) produces a U-shaped curve in prediction accuracy for the following reasons:
> * Small number of levels: Too many mesh nodes are aggregated into the same cell, blurring local distinctions and increasing error.
> * Large number of levels: Each cell contains only a single node, preventing tokens from capturing interactions between neighboring nodes. This again degrades accuracy despite the higher resolution.
>
> For dtoken, model accuracy improves as the number of latent tokens increases, but the training epoch time rises noticeably once a certain threshold is exceeded. In general, a typical choice for datasets fewer than 10k mesh nodes is 512 latent tokens, which provides sufficient capacity while maintaining comparable training and inference efficiency to smaller values.

---

> ### Author Response · Authors · 2025-09-25
>
> **5. Add a comparison of accuracy and efficiency with commercial FEM software on the same problem.**
>
> We appreciate the reviewer’s suggestion. However, we view commercial FEM solvers as the ground truth for our experiments—our simulations are generated directly using FEM. Therefore, an accuracy comparison against FEM itself would not be meaningful, since our method is designed to approximate those solutions as closely as possible.
>
> Regarding efficiency, our work primarily focuses on comparing learning-based approaches within a unified framework. In Section 5.4, we report both memory and runtime efficiency relative to existing learning-based methods.
> For comparisons with CPU-based FEM solvers, we refer the reviewer to prior work such as MGN[1], where the authors benchmarked surrogate model inference speed against commercial FEM solvers. They showed that when run on GPU, the learned surrogate achieved a 20–200$\times$ speedup depending on the problem setting (see Section A.5 of [1] for details).
> Since our work already benchmarks efficiency against representative approaches such as MGN, we did not expand our study to include direct comparisons with FEM solvers.
>
> [1] Pfaff, T., Fortunato, M., Sanchez-Gonzalez, A., & Battaglia, P. (2020, October). Learning mesh-based simulation with graph networks. In International conference on learning representations.
>
> **6. Explore and discuss the scenarios and limiting conditions under which the method might encounter failures.**
>
> We added a limitation analysis in the Limitation and Future Work (Section 6) to better acknowledge interested readers on the limitations of our methods: “Despite its effectiveness, the method has certain limitations. Specifically, the octree structure enforces a fixed set of cell lengths across levels, making it difficult to identify an optimal configuration. As shown in Figure 7, the performance curve remains relatively flat near the optimal $L_{cell}$, but a more flexible and finer grid control could yield further improvements.”
>
> **7. Commit to making the code and the ABCD-XL dataset publicly available upon acceptance. Open-sourcing the code and dataset will significantly enhance the reproducibility of the research.**
>
> Yes, we are actively working with the relevant parties in our institution to make the ABCD-XL dataset and the code publicly available. The internal approval process including the auditing is expected to take several weeks or even longer to complete. Once finalized, we will release the dataset and code repository to public.

---

> > ### Comment · Reviewer_Zaye · 2025-10-13
> > **Official Comment by Reviewer Zaye**
> >
> > I deeply appreciate the authors' painstaking revision of the manuscript. The key concerns I put forward previously have been appropriately dealt with in the present version. In particular, the stability analysis has been thoroughly discussed, and a more elaborate introduction to the proposed ABCD - XL dataset has been furnished. In my view, this article has made valuable contributions. Once the code and the ABCD - XL dataset are made publicly accessible, it will attract the attention of the ML community. Consequently, I recommend the acceptance of this manuscript.

---

### Review · Reviewer_UfKQ · 2025-09-13

**Summary Of Contributions:**

Authors propose an architecture for learning surrogate models of deformable bodies interaction. The main components of the approach are:
1. GNN used as part of encoder
2. Special procedure that transfer features from mesh to features on multilevel structure grid
3. Sparse convolution on multilevel structured grid (for one dataset)
4. Downsampling of the grid with Farthest Point Sampling algorithm
5. Attention blocks of encoder and processor
6. Decoder that transfer features back from structured grid to heterogeneous graph where physical observables are defined

The resulting architecture is tested on several benchmarks and is reported to have better accuracy and scalability when compared with baselines.

**Audience:**

Yes

**Audience Explanation:**

The paper presents surrogate modelling for an applied problem in continuum mechanics. It may be of interest for AI4Science community that develop neural solvers for partial differential equations.

**Claims And Evidence:**

Yes

**Claims Explanation:**

The main claims are supported by empirical evaluation.

**Requested Changes:**

1. Authors write "However, FEM typically incurs high computational costs and requires significant manual effort from engineers to ensure solver stability and convergence" and reference https://arxiv.org/abs/2211.00713. Is it a reliable source for such claim? The article mentioned is on deep learning, so the authors of this article have incentive to motivate their research. Can the authors please consider backing up high costs of FEMs with some other source, or numerical test, or estimation?
2. The idea to represent irregular data by projecting on regular grid appears in many publications. For examples in generative AI see https://arxiv.org/abs/2306.07473 and references therein. The oldest example from scientific computing known to me is https://ieeexplore.ieee.org/abstract/document/662670.

   I suggest authors discuss prior art and include relevant references (not necessarily mentioned above) to improve the presentation of their ideas.
3. For the description of Farthest Point Sampling authors reference https://arxiv.org/abs/1706.02413. Is it an appropriate reference? FPS was not introduced in this paper. I suggest authors to cite original sources of FPS, e.g., https://ieeexplore.ieee.org/document/623193.
4. Farthest Point Sampling is an essential part of the architecture that ensures scalability. Anchor Attention described in https://arxiv.org/abs/2505.16463 seems to be related technique. Can the authors discuss difference between FPS and Anchor Attention?
5. In Table 1 authors report RMSE for stress and displacement. Can the authors provide reference values of strain and displacement for the simulations from considered datasets (i.e., what is the order of magnitude for these quantities)? Alternatively, authors may want to provide relative error in place of absolute ones. Besides that, both stress and displacement are not dimensionless, so some form of normalisation is required.
6. Figure 9 shows RMSE for different number of levels in multilevel structured grid. Can the authors explain why it is a U-shape curve? It looks slightly counterintuitive to me that accuracy deteriorates for large $L$.
7. The other interesting hyperparameter is the number of latent tokens. Can the authors report results on how accuracy depends on the number of latent tokens? What is typical number of latent tokens for each dataset?
8. In section 4.2.4 the latent tokens are selected based on FPS algorithm. Can the authors provide an illustration of points selected by FPS?
9. Can the authors share details how FPS is implemented?

---

> ### Author Response · Authors · 2025-09-25
>
> We appreciate the reviewer’s thorough and insightful comments, which have greatly strengthened the clarity and rigor of our work.
>
> **1. [question omitted to save space]**
>
> Thank you for pointing out the potential bias in the original reference. We have updated the citations to include established FEM domain literature that more solidly supports our statement [1][2][3]. These works highlight the high computational costs and significant manual efforts often required to ensure solver stability and convergence in FEM. In addition, we added a recent dataset paper [4], where the authors report CPU hours and memory usage for large-scale car CFD simulations using FEM, providing a concrete example of the computational burden of classical FEM methods. Reference change has been made in section 1 correspondingly.
>
> [1] Hughes, T. J. (2003). The finite element method: linear static and dynamic finite element analysis. Courier Corporation.
>
> [2] O. Zienkiewicz (2000). The finite element method. The Basis, Vol. 1.
>
> [3] Cook, R. D. (2007). Concepts and applications of finite element analysis. John Wiley & Sons.
>
> [4] Elrefaie, M., Morar, F., Dai, A., & Ahmed, F. (2024). Drivaernet++: A large-scale multimodal car dataset with computational fluid dynamics simulations and deep learning benchmarks. Advances in Neural Information Processing Systems, 37, 499–536.
>
> **2. The idea to represent irregular data by projecting on regular grid appears in many publications. For examples in generative AI see [link] and references therein. The oldest example from scientific computing known to me is [link]. I suggest authors discuss prior art and include relevant references (not necessarily mentioned above) to improve the presentation of their ideas.**
>
> Thanks for raising this point, indeed, projecting data from irregular mesh to regular grid is worth discussion to better present our idea. We added following reference and discussion in section 1:
>
> Thuerey et al. [1] interpolated the input 2D field onto a 128×128 grid and applied a convolutional U-Net to process the grid input for CFD simulation. Pinheiro et al. [2] proposed representing atoms as continuous densities and molecules as discretizations of 3D space on voxel grids. Similar to image tasks, they applied a score-based generative model to generate molecules by denoising noisy voxelized representations. Beyond regular grids suitable for convolutional operations, later methods such as GraphCast (Lam et al., 2023) introduced a background grid to which the mesh is attached, enabling more efficient message passing in graph neural networks. However, this effectively converts an unstructured mesh into a structured one, which may work well for simple, regular geometries but can suffer from reduced accuracy and efficiency when applied to complex shapes.
>
> **3. For the description of Farthest Point Sampling authors reference [link]. Is it an appropriate reference? FPS was not introduced in this paper. I suggest authors to cite original sources of FPS.**
>
> The TIP ’97 paper (Eldar et al., 1997) is indeed the canonical reference for the farthest point sampling strategy in signal processing, while the NeurIPS 2017 paper (Qi et al., 2017) established FPS as the standard sampling method for point sets in 3D vision. We have updated our references to include both works for a more complete attribution.
>
> **4. Farthest Point Sampling is an essential part of the architecture that ensures scalability. Anchor Attention described in [link] seems to be related technique. Can the authors discuss difference between FPS and Anchor Attention?**
>
> Instead of aggregating tokens into a subset, Anchor Attention approximates full attention by introducing a set of anchor tokens. This provides a general mechanism for reducing the cost of full attention to a level comparable to cross attention, making it a promising alternative to our FPS + CA strategy. However, Anchor Attention places emphasis on learning anchor tokens to approximate full attention, whereas our design prioritizes retaining the spatial fidelity of tokens through structured sampling. Due to this difference in design philosophy, the two approaches are not directly compatible. After careful consideration, we chose to focus our contribution within the literature on projecting irregular data onto regular grids, and we have revised the related works section accordingly. Nevertheless, exploring full-attention approximations remains an interesting direction, and we have highlighted it as future work in the added Section 6: “In addition, we employed FPS and CA to construct an efficiency-oriented bottleneck, yet recent work such as AnchorFormer~\cite{shan2025anchorformer} introduces anchor-based approximations for full attention, offering a promising alternative that can preserve efficiency while potentially improving accuracy.”

---

> ### Author Response · Authors · 2025-09-25
>
> **5.1 In Table 1 authors report RMSE for stress and displacement. Can the authors provide reference values of strain and displacement for the simulations from considered datasets (i.e., what is the order of magnitude for these quantities)?**
>
> For the DeformingPlate dataset, the calculated values are:
> * Stress (max-principal stress): mean = 25,545, variance = 50,515
> * Displacement: mean = [0.0049, –0.0009, 0.0035], variance = [0.0100, 0.0060, 0.0274]
>
> For the ABCD dataset, the values are:
> * Stress (stress tensor in Voigt notation): mean = [-3.0967e-01, -2.8043e-02, -2.4857e-02, 6.5793e-04, -1.5009e-03,  1.4313e-04], variance = [2.2231, 0.8907, 0.9410, 0.4715, 0.5249, 0.2212]
> * Displacement: mean = [–9.1e–04, –1.6e–05, –5.9e–05], variance = [0.0035, 0.0029, 0.0032]
>
> While the exact units of stress and displacement were not explicitly reported in the original MeshGraphNets paper, we verified that for the ABCD dataset the units are MPa ($N$/$mm^2$) for stress and meters ($m$) for displacement. As finite element analysis (FEA) itself has no inherent length scale and the prior datasets did not consistently report this information, we chose not to impose additional assumptions for alignment across datasets.
>
> **5.2 Alternatively, authors may want to provide relative error in place of absolute ones.**
>
> As suggested, we added an additional evaluation using Mean Absolute Percentage Error (MAPE) in Section B.1. MAPE expresses prediction error as a percentage and is scale-independent, making it suitable for relative comparisons. However, it is unstable when ground-truth values approach zero, which is frequently the case for DeformingPlate and ABCD. Therefore, RMSE remains the primary and more reliable metric for this domain, consistent with prior work.
>
> **5.3 Besides that, both stress and displacement are not dimensionless, so some form of normalisation is required.**
>
> We agree with the reviewer that normalization is crucial for stable training. This was indeed applied but not explicitly documented in the original submission. We have now clarified this in Section A.2 of the Appendix by adding: “The input and target features are normalized to zero mean and unit variance based on the statistics of the training set.”
>
> **6. Figure 9 shows RMSE for different number of levels in multilevel structured grid. Can the authors explain why it is a U-shape curve? It looks slightly counterintuitive to me that accuracy deteriorates for large L.**
>
> Thank you for the insightful question. The U-shaped curve in Figure 9 (Figure 7 in the revised manuscript) reflects the core idea of our method: quantization cells are essential for encoding interactions between mesh nodes.
> * Small number of levels: Too many mesh nodes are mapped into the same cell, which leads to reduced resolution and leads to higher errors.
> * Large number of levels: Cells become so small that each contains only one or very few nodes; “tokens then fail to capture interactions among neighboring nodes, degrading accuracy despite the finer resolution.”
>
> Thus, intermediate quantization levels strike the right balance, preserving local interactions while avoiding excessive aggregation, and leads to the lowest RMSE. We have added the quoted line above to our manuscript to improve clarity.
>
> **7. The other interesting hyperparameter is the number of latent tokens. Can the authors report results on how accuracy depends on the number of latent tokens? What is typical number of latent tokens for each dataset?**
>
> We added an analysis of the number of latent tokens in Section 5.3, under the paragraph Number of Latent Tokens. In general, model accuracy improves as the number of latent tokens increases, but the training epoch time rises noticeably once a certain threshold is exceeded. A typical choice is 512 latent tokens, which provides sufficient capacity for simulation datasets with fewer than 10k mesh nodes, while maintaining comparable training and inference efficiency to smaller values.
>
> **8. In section 4.2.4 the latent tokens are selected based on FPS algorithm. Can the authors provide an illustration of points selected by FPS?**
>
> We revised Figure 2 to include this illustration. In the cell diagram before FPS, the points selected by the FPS algorithm are highlighted in green. This illustration is intended to provide a clearer sense of which points are chosen by FPS and subsequently processed in the bottleneck.
>
> **9. Can the authors share details how FPS is implemented?**
>
> We have added an algorithm diagram in Section A.6. Since FPS is a standard operation in point cloud processing, our implementation directly relies on the version provided in the torch_cluster library.

---

> > ### Comment · Reviewer_UfKQ · 2025-10-12
> >
> > I would like to thank the authors for the thoughtful revision of the manuscript. My main concerns are addressed in the current version: related literature is sufficiently discussed, additional data on experiments is available. In my view an article will be of interest for the part of the ML community interested in scientific computing problems, so my recommendation is to accept the manuscript.

---

### Review · Reviewer_k1UN · 2025-09-13

**Summary Of Contributions:**

The paper proposed a novel Adaptive Spatial Tokenization method for efficient representation of physical states, overcoming the challenges of computationally intensive and impractical large-scale meshes required by pairwise global edges. It then developed a transformer-based model with both cross-attention and self-attention that operates on these spatial tokens in the latent space. The proposed method has been demonstrated to outperform state-of-the-art baselines on both two public datasets and a newly constructed dataset, not only in accuracy but also in efficiency and scalability.

**Audience:**

Yes

**Audience Explanation:**

From the perspective of a non-expert, the proposed algorithm outperforms several recent methods on public datasets in terms of accuracy, efficiency, and scalability. In addition, the paper introduces a new dataset that may be of interest to researchers in the same field.

**Broader Impact Concerns:**

N\A

**Claims And Evidence:**

Yes

**Claims Explanation:**

Comparing with three baselines proposed in 2020, 2023, and 2024 respectively, which are claimed to be state-of-the-art, the proposed algorithm achieves the best performance on displacement and stress. Furthermore, its computational efficiency and scaling capability are also impressive. These points provide evidence supporting my recommendation, though I am a non-expert.

However, there are also some concerns:

1. The novelty is not entirely clear. Although AST is presented as a novel method, it is unclear which specific component is novel. Moreover, how the learnable tokenization achieves higher accuracy while also overcoming computational intensity is not analyzed in detail.

2. The evaluation metrics in table 1 are not well explained. Additionally, how should the visualization results in Figures 6 and 7 be interpreted? What signals indicate a better prediction in those visualizations?

3. Since the proposed algorithm also uses the same background grid as in GraphCast that converts an unstructured mesh into a structured one, why does GraphCast suffer from reduced accuracy while the proposed algorithm does not? Additionally, could GraphCast serve as a competitive baseline as well?

4. In the paper by Pfaff et al. (2020) introducing GraphMeshNets, I did not find any results regarding the dataset named SphereSimple. Did I miss something? Furthermore, why was GraphMeshNets not included as a baseline? I am also unclear about the evaluation metrics used in Pfaff et al. (2020) compared to those in the current paper. Did both papers use the same set of metrics? For the DeformingPlate dataset in Pfaff et al. (2020), it appears they reported only a single metric for rollout-all, with a value of 15.1 ± 4.0, whereas the current paper reports two metrics: one for displacement and one for stress. Could you clarify the differences between these two sets of metrics and how they can be made comparable?

**Requested Changes:**

Please refer to the previous section for explanations regarding novelty, evaluation metrics, and other related points.

---

> ### Author Response · Authors · 2025-09-25
>
> Thank you for carefully pointing out these details. We agree that clarification is warranted.
>
> **1.1 The novelty is not entirely clear. Although AST is presented as a novel method, it is unclear which specific component is novel.**
>
> The novelty for the proposed AST methods could be summarized as below:
> * Compact and fixed-length representation for physics field in 3D: In the proposed AST methods, we created an encoder to map unstructured meshes into structured grid cells and then compresses them into a fixed-length representation through cross-attention.
> * A representation that models the interaction implicitly: Unlike prior GNN-based approaches that require dynamic pairwise edges for all interacting nodes, AST avoids explicit pairwise connectivity construction.
> * Ready for attention based operations: By encode the physics states into a fixed length representation, it is better suited for attention based operations comparing with raw graph inputs.
>
> The contributions in the introduction were revised accordingly.
>
> **1.2 Moreover, how the learnable tokenization achieves higher accuracy while also overcoming computational intensity is not analyzed in detail.**
>
> Previously, Graph Neural Network (GNN)–based methods relied on calculating pairwise distances between nodes and constructing dynamic edges to model interactions. This process typically scales as $O(N^2)$, where N is the number of mesh nodes, and quickly becomes intractable for large-scale meshes. In contrast, AST maps mesh nodes into $M$ tokens, where $M \ll N$, and processes interactions globally through self-attention on these tokens. For example, meshes from the ABCD-XL dataset typically contain ~100,000 nodes. GNN-based methods such as MGN become intractable at this scale. In AST, we map nodes into 512 tokens (each of dimension 128), enabling downstream prediction with tractable complexity and strong performance. Scaling and accuracy comparisons are provided in Figures 9 and 10.
>
> **2. The evaluation metrics in table 1 are not well explained. Additionally, how should the visualization results in Figures 6 and 7 be interpreted? What signals indicate a better prediction in those visualizations?**
>
> We clarify that the reported values are root mean square error (RMSE) between predicted and ground-truth rollout trajectories:
> - Displacement error ($\mathbf{u}$): $\mathbf{u} = \mathbf{x}_t - \mathbf{x}_0$ denotes node displacements. RMSE is measured across all nodes and rollout steps, scaled by $10^{-3}$ for comparability.
> - Stress error ($\sigma$): RMSE of von Mises stress across all nodes and rollout steps.
>
> All results are reported as mean ± standard deviation over 3 independent runs. “OOM” denotes out-of-memory and “Diverge” indicates rollout instability. We added the clarification “RMSE is computed as described in Section A.1, with results shown as mean ± standard deviation over three independent runs.” to the caption of Table 1.
>
> For the figures, we revised captions to improve interpretability:
> - Figure 6: “Warmer colors correspond to higher feature norms. Their concentration at collision regions indicates that spatial cells effectively capture critical interactions between disconnected meshes.”
> - Figure 7 (now Figure 11): “Our method yields predictions that are closer to the ground truth at the collision region.”
> - Figure 8 (now Figure 12): “Our method produces shapes that more closely match the ground truth, indicating more accurate displacement predictions. Red-colored cells highlight correspondingly closer stress predictions.”

---

> ### Author Response · Authors · 2025-09-26
>
> **3. Since the proposed algorithm also uses the same background grid as in GraphCast that converts an unstructured mesh into a structured one, why does GraphCast suffer from reduced accuracy while the proposed algorithm does not? Additionally, could GraphCast serve as a competitive baseline as well?**
>
> Although GraphCast also converts unstructured meshes into structured grids, it differs significantly from AST:
> - Octree for multi-resolution: We map mesh nodes to a uniform spatial grid, then encode them into an Octree that refines only occupied cells, improving accuracy for complex geometries. While GraphCast leverages the natural geometry of a sphere to construct its background grid with abundant long-range edges, extending this approach to more general shapes requires additional work.
> - Cross-attention for learnable mapping: Mesh nodes are encoded into spatial tokens via the Octree, then further compressed into a compact fixed-length representation using cross-attention. These choices make AST more flexible and better suited for attention-based architectures. In GraphCast, the raw data are projected onto the background grid which facilitates message-passing(MP) for distant nodes, but the processor itself remains a stack of MP layers. Since our experiments already include MGN and BSMS as representative MP baselines, adding GraphCast would be redundant.
>
> We added GraphCast into our literature study: "later methods such as GraphCast (Lam et al., 2023) introduced a background grid to which the mesh is attached, enabling more efficient message passing in graph neural networks. However, this effectively converts an unstructured mesh into a structured one, which may work well for simple, regular geometries but can suffer from reduced accuracy and efficiency when applied to complex shapes."
>
> **4.1 In the paper by Pfaff et al. (2020) introducing GraphMeshNets, I did not find any results regarding the dataset named SphereSimple. Did I miss something?**
>
> The MeshGraphNets paper did not report results on SphereSimple. This dataset was only included in their codebase. In the paper, results were presented on SphereDynamic, which uses re-meshing at each rollout step. Our work focuses on fixed-mesh settings in FEA simulations, making SphereSimple the more relevant choice. This aligns with HCMT [https://arxiv.org/pdf/2312.12467], which also evaluates on SphereSimple.
>
> **4.2 Furthermore, why was GraphMeshNets not included as a baseline?**
>
> MeshGraphNets is included as a baseline in our paper under the abbreviation MGN.
>
> **4.3 I am also unclear about the evaluation metrics used in Pfaff et al. (2020) compared to those in the current paper. Did both papers use the same set of metrics? For the DeformingPlate dataset in Pfaff et al. (2020), it appears they reported only a single metric for rollout-all, with a value of 15.1 ± 4.0, whereas the current paper reports two metrics: one for displacement and one for stress. Could you clarify the differences between these two sets of metrics and how they can be made comparable?**
>
> In Table 1 of Pfaff et al. (2020), only displacement RMSE was reported. However, the DeformingPlate dataset has two outputs: displacement and stress (see Appendix A.1, “hyperelastic” system). More recent works (e.g., HCMT, Table 2) report both displacement and stress errors, which we also follow.
>
> Regarding performance differences, our reimplementation of MGN achieves significantly lower displacement error (5.5 ± 0.2 vs. 15.1 ± 4.0 in Pfaff et al., 2020). We attribute this improvement to the improved training setup: The original paper trained with a batch size of 1 and exponential learning rate decay over 5M steps (~25 epochs), due to limited compute. In our experiments, we trained with batch size 48, 100 epochs, and a cosine learning rate schedule with warmup (details in Section A.2).

---

### Decision · Action_Editor_SXYw · 2025-11-02

**Recommendation:** Accept as is

**Audience:**

Yes

**Audience Explanation:**

The paper develops a surrogate model for a continuum mechanics problem, positioning it as relevant to the AI4Science and neural PDE solver communities. Its primary contributions are practical, offering a novel solution that enables industrial-grade simulation. The work is highlighted for its direct application potential in fields like materials science and robotics, its computational efficiency improvements, and its specific capability to bridge a gap in simulating complex, large-scale deformable body interactions.

**Claims And Evidence:**

Yes

**Claims Explanation:**

The main claims of the paper are supported comprehensive quantitative evaluation with strong baseline methods on four datasets in terms of RMSE, efficiency, and scaling capability.